

# Biomass burning plume chemistry: OH radical initiated oxidation of 3-penten-2-one and its main oxidation product 2-hydroxypropanal

Niklas Illmann[1], Iulia Patroescu-Klotz[1], and Peter Wiesen[1]

[1] Institute for Environmental and Atmospheric Research, Bergische Universität Wuppertal, Gaußstr. 20, 42097 Wuppertal, Germany

*Correspondence to*: Niklas Illmann (illmann@uni-wuppertal.de)

**Abstract.** In order to enlarge our understanding of biomass burning plume chemistry, the OH radical initiated oxidation of 3-penten-2-one (3P2), identified in biomass burning emissions, and 2-hydroxypropanal (2HPr) were investigated at $298 \pm 3$ K and $990 \pm 15$ mbar in two atmospheric simulation chambers using long-path FTIR spectroscopy. The rate coefficient of 3P2 + OH was determined to be $(6.2 \pm 1.0) \times 10^{-11}$ cm$^3$ molecule$^{-1}$ s$^{-1}$ and the molar first-generation yields for acetaldehyde, methyl glyoxal, 2HPr and the sum of PAN and CO$_2$, used to determine the CH$_3$C(O) radical yield, were $0.39 \pm 0.07$, $0.32 \pm 0.08$, $0.68 \pm 0.27$, and $0.56 \pm 0.14$, respectively, under conditions where the 3P2 derived peroxy radicals react solely with NO. The 2HPr + OH reaction was investigated using 3P2 + OH as a source of the α-hydroxyaldehyde adjusting the experimental conditions to shift the reaction system towards secondary oxidation processes. The rate coefficient was estimated to be $(2.2 \pm 0.6) \times 10^{-11}$ cm$^3$ molecule$^{-1}$ s$^{-1}$. Employing an Euler-Cauchy model to describe the temporal behaviour of the experiments, the further oxidation of 2HPr was shown to form methyl glyoxal, acetaldehyde and CO$_2$ with estimated yields of $0.21 \pm 0.05$, $0.79 \pm 0.05$, and $0.79 \pm 0.05$, respectively.





## 1 Introduction

Unsaturated ketones rise an increasing interest as more sources for their atmospheric burden are uncovered. They are potentially formed in the atmospheric oxidation of terpenes involving OH radicals or $O_3$ molecules. 3-Buten-2-one (= methyl vinyl ketone) is the most famous representative formed through the gas-phase oxidation of the most abundant NMHC, namely isoprene (Calvert et al., 2000). Other $\alpha,\beta$-unsaturated ketones, however, are used in the food and fragrances industry (Bickers et al., 2003). Ciccioli et al. (2001) investigated controlled biomass burning emissions representative for the Mediterranean vegetation and identified 3-penten-2-one (3P2) in both the flaming and smoldering of pine wood. Hatch et al. (2017) identified 3P2 among various oxygenated species in the gas-phase biomass burning emissions of Chinese rice straw, Indonesian peat and boughs of ponderosa pine and black spruce during the FLAME-4 campaign. Besides, they also identified 2-methylpenta-1,3-diene (Hatch et al., 2017) which potentially yields 3P2 in the further gas-phase oxidation. All these findings pinpoint 3P2 as a likely constituent of most biomass burning (BB) plumes.

Understanding the BB plume chemistry is of global interest since their photochemical aging is believed as a potentially significant contributor to ozone and organic aerosol formation (Jaffe et al., 2008; Yokelson et al., 2009; Alvarado et al., 2015). For instance, Jaffe et al. (2008) found a correlation between the interannual variation of $O_3$ and burned area in the western USA. During the next decades a drier climate, expected due to global warming, will likely result in an increase of both the number and intensity of fire events worldwide, which could consequently enlarge the influence of biomass burning on air quality. In some field measurement campaigns of BB plumes have shown that wildfires may increase significantly the ozone enhancement ratios ($\Delta O_3/\Delta CO$) (Mauzerall et al., 1998; Honrath et al., 2004) while in others a correlation between $O_3$ and CO could not be observed (Alvarado et al., 2010). Given typical VOC/$NO_x$ emission ratios from biomass burning, ozone formation is mainly limited by the availability of $NO_x$, which, in turn, depends on the fuel nitrogen content and the combustion efficiency (Jaffe and Wigder, 2012 and references therein). Alvarado et al. (2010) were the first to observe a fast peroxyacetyl nitrate (PAN) production in a young boreal smoke plume within the first hours after emission. Simulations of a young BB plume from a prescribed fire showed the evolution of secondary organic aerosol and $O_3$ to be sensitive to unidentified VOCs whose chemistry is likely characterized, amongst others, through (a) OH rate coefficients in the order of $10^{-11}$ cm$^3$ molecule$^{-1}$ s$^{-1}$, (b) $RO_2$ + NO reactions resulting mainly in fragmentation, and (c) an efficient $HO_2$ regeneration (Alvarado et al., 2015). However, this resulted still in a significant overestimation of downwind PAN formation (Alvarado et al., 2015). Accordingly, given the complexity of biomass burning smoke and the various conditions possible within both a young and an aged plume the chemistry is still not well characterized. Besides, the oxidation of very reactive organic species seems to be crucial for a comprehensive picture.

The present work therefore contributes to expand our understanding of BB plume chemistry by studying the oxidation processes of single species, identified in gas-phase emissions, in simulation chamber experiments. In this respect we investigated the OH radical initiated oxidation of 3P2 and the fate of its main oxidation product. Up to now, to the best of our





knowledge, only kinetic data were reported for the reactions of 3P2 with OH radicals and Cl atoms (Blanco et al., 2012), NO$_3$

radicals (Canosa-Mas et al., 2005) and O$_3$ (Greene and Atkinson, 1994; Sato et al., 2004; Illmann et al., 2021a).

## 2. Experimental

Kinetic and product study experiments were conducted in two indoor simulation chambers at 298 ± 3 K and in 990 ± 15 mbar

of synthetic air. In both chambers OH radicals were generated by the photolysis of methyl nitrite in the presence of sufficient

amounts of NO to suppress any ozone formation and consequently the generation of NO$_3$ radicals. Methyl nitrite has been

synthesized by the dropwise addition of sulphuric acid to a saturated aqueous solution of sodium nitrite in methanol according

to a method previously outlined by Taylor et al. (1980). The product was collected and stored in a cooling trap at 195 K. Its

purity was verified via FTIR spectroscopy.

### 2.1 480 L chamber

The cylindrical borosilicate glass tube with a length of 3 m and 0.45 m inner diameter is surrounded by 32 superactinic

fluorescent lamps (Philips TL A 40W: 300–460 nm, I$_{max}$ = 360 nm) and closed at both ends by aluminium flanges. These

contain various ports for the introduction of reactants and bath gas and the coupling with analytical devices. The pumping

system consists of a rotary vane pump and a roots pump yielding an end vacuum up to 10$^{-3}$ mbar. A White-type mirror system

is installed inside the chamber (optical path length: 50.4 ± 0.2 m) and coupled to a Nicolet 6700 FTIR spectrometer to monitor

reactants and products. Spectra are recorded in the spectral range 4000–700 cm$^{-1}$ with a resolution of 1 cm$^{-1}$. The present set-

up of the chamber is described with further details in the recent literature (Illmann et al., 2021b).

The initial mixing ratios in the 480 L chamber experiments in ppmV (1 ppmV = 2.46 × 10$^{13}$ molecules cm$^{-3}$ at 298 K)

were: 5.8–9.4 for 3P2, 5.0 for isoprene, 8.3 for E2-butene, 10–16 for methyl nitrite, and 20–27 for NO.


### 2.2 1080 L chamber

The 1080 L chamber consists of two joint quartz-glass tubes with a total length of 6.2 m and 0.47 m inner diameter. It is

surrounded by 32 superactinic fluorescent lamps (Philips TL05 40W: 300–460 nm, I$_{max}$ = 360 nm) and 32 low-pressure

mercury vapour lamps (Philips TUV 40W, I$_{max}$ = 254 nm) which can be switched individually. The pumping system consists

of a turbo-molecular pump backed by a double-stage rotary fore pump to yield an end vacuum of 10$^{-4}$ mbar. The White-type

mirror system installed inside the chamber is operated at a total optical path length of 484.7 ± 0.8 m and coupled to a Nicolet

iS50 FTIR spectrometer recording FTIR spectra in the range 4000–700 cm$^{-1}$ with a resolution of 1 cm$^{-1}$. A more detailed

description of the chamber can be found in the recent literature (Illmann et al., 2021b).





The initial mixing ratios in the 1080 L chamber experiments, in ppmV (1 ppmV = $2.46 \times 10^{13}$ molecules cm$^{-3}$ at
298 K), were: 1.1–1.3 for 3P2, 0.9–1.1 for isoprene, 1.3–1.5 for E2-butene, 0.9–1.9 for methyl nitrite, and 2.0–3.7 for NO,
1.5–1.7 for 3-buten-2-ol, 1.3–1.4 for 3-penten-2-ol, and 13000–17000 for CO.

## 2.3 Methods

The rate coefficient of the 3P2 + OH reaction has been determined using the relative-rate technique, thus by relating the
consumption of 3-penten-2-one to the consumption of a reference compound. If reactions other than presented below (R1–R3)
are negligible in the experimental set-up,

3P2 + OH → products                                                                                (R1)
3P2 + wall → loss                                                                                  (R2)
reference + OH → products                                                                           (R3)

the following equation can be used to determine the rate coefficient $k_{3P2}$:

$$\ln\left(\frac{[3P2]_0}{[3P2]_t}\right) - k_{\text{loss}} \times t = \frac{k_{3P2}}{k_{ref.}} \times \ln\left(\frac{[\text{ref.}]_0}{[\text{ref.}]_t}\right) \tag{1}$$


where $[X]_t$ is the concentration of the species X at time t. The rate coefficient ratio $k_{3P2}/k_{ref}$ used to calculate $k_{3P2}$ were thus
obtained from regression analysis after plotting $\left\{\ln\left(\frac{[3P2]_0}{[3P2]_t}\right) - k_{\text{loss}} \times t\right\}$ against $\left\{\ln\left(\frac{[\text{ref.}]_0}{[\text{ref.}]_t}\right)\right\}$.

Mixing ratios of identified species in the product study experiments were obtained by subtracting calibrated reference
FTIR spectra of the target species. The cross sections we used for calibration were taken either from the literature, in the case
of methyl glyoxal (Profeta et al., 2011; Talukdar et al., 2011) and peroxyacetyl nitrate (Allen et al., 2005), the internal
Wuppertal laboratory database (acetaldehyde) or determined within this work (3-penten-2-one, 2-hydroxypropanal). $CO_2$ was
quantified by the integration of the absorption features in the spectral range 2400–2349 cm$^{-1}$ and a polynomial calibration
function derived from the injection of various volumes of $CO_2$ using a calibrated gas-tight syringe. Uncorrected molar
formation yields of the reaction products were calculated by plotting the mixing ratio of formed product against the mixing
ratio of the consumed 3-penten-2-one. These yields were corrected for secondary reactions in the experimental set-up using
the Euler-Cauchy approach previously outlined in the recent literature (Illmann et al., 2021b). Here, the differential equations
are constructed based on the simplified reaction sequence of each species. The molar formation yields for products of the target
reaction are included as a parameter to be varied until the simulated temporal behaviour of each species matches the
experimental data.





Cross sections and reference FTIR spectra of 2HPr were obtained by the in situ generation of the aldehyde through the ozonolysis of 3-buten-2-ol and 3-penten-2-ol, respectively, in the presence of sufficient CO to scavenge any OH radical formed during the $O_3$ reaction, in the 1080 L chamber. In order to investigate the OH oxidation of 2HPr, the 3P2 + OH reaction was used as a source of the α-hydroxyaldehyde. An estimation of the rate coefficient was obtained following the procedure outlined previously by Baker et al. (2004). Branching ratios for the product formation were derived from modelling the

temporal behaviour of the relevant species using our recently presented approach (Illmann et al., 2021b).

Typically, 15 spectra were recorded per experiment and the first five spectra were collected in the dark to determine potential wall losses in each experiment. In the product study experiments additional 5 spectra were recorded after the OH reaction was terminated (switch-off the lamps) in order to check for the wall loss rates of each formed reaction product. For the 2HPr + OH investigations, a second methyl nitrite and NO injection occured after the first irradiation period, in the dark,

followed by further irradiation. About 50–70 scans were co-added per spectrum which leads to averaging periods of about 80–115 s. Time intervals for both irradiation and dark periods of the experiments were in the order of 15 min.

For economic reasons, the housing which infolds the transfer optics between FTIR spectrometer and chamber is flushed with purified dry air. Therefore, quantification of $CO_2$ is, due to a slight variability in the dry air supply, unreliable under normal laboratory conditions. To be able to quantify $CO_2$, in selected product study experiments, the transfer optics

housing was flushed with ultrapure $N_2$ evaporated from a liquid nitrogen tank.

## 2.4 Materials

The following chemicals have been used without further purification and purities as stated by the suppliers: 3-penten-2-one (Alfa Aesar, technical grade 85%), trans-2-butene (Messer, 99%), isoprene (Aldrich, 99%), 3-buten-2-ol (Alfa Aesar, 97%),

3-penten-2-ol (Sigma Aldrich, 96%), NO (Air Liquide, 99.5%), CO (Air Liquide, 99.97%), synthetic air (Messer, 99.9999%). While the supplier states the predominance of the trans isomer for 3-penten-2-ol the cis/trans isomer ratio is not specified for 3-penten-2-one. The latter compound is not commercially available with purities higher than 85%. Another sample of 70% purity (technical grade) contains, as stated by the supplier, mainly 4-methyl-3-penten-2-one as impurity. However, the gas-phase FTIR spectra of the 70% sample after subtraction of the 4-methyl-3-penten-2-one content are identical to the spectra

recorded for the 85% sample. It is therefore reasonable to assume that no absorptions other than those belonging to 3-penten-2-one are present in the spectra of the used sample.



# 3. Results and discussion

The first-order wall loss of 3P2 was $< 1 \times 10^{-5}$ s$^{-1}$ in all 480 L chamber experiments and in the range of $(5 - 10) \times 10^{-5}$ s$^{-1}$ in
the 1080 L chamber experiments, respectively. Photolysis and dark reactions between 3P2 and the radical source were found
to be negligible under all experimental conditions.

## 3.1 3-Penten-2-one + OH kinetics

Kinetic experiments were performed in both chambers under varying light intensity using isoprene and E2-butene as
references. The relative-rate plots according to eq. (1) are presented in Figure 1 for all performed experiments. The relative
ratios $k_{3P2}/k_{ref}$, determined for each individual experiment following regression analysis, agree within $< 13\%$ using isoprene
and $< 9\%$ using E2-butene as reference compound, respectively. Intercepts of the regression lines were found to be zero within
a 2σ statistical error and the correlation coefficients were $R^2 > 0.99$. Given the latest IUPAC recommendations (Mellouki et
al., 2021) for the rate coefficients of the OH radical reactions with E2-butene ($k = (7.1 \pm 1.1) \times 10^{-11}$ cm$^3$ molecule$^{-1}$ s$^{-1}$) and
isoprene ($k = (1.0 \pm 0.2) \times 10^{-10}$ cm$^3$ molecule$^{-1}$ s$^{-1}$), respectively, the calculated rate constants for 3P2 derived from both
references are in excellent agreement. All this suggests that secondary processes other than wall loss can be neglected in the
present experimental set-up. Table 1 summarises the whole experimental results. The weighted average rate coefficient is
$(6.2 \pm 1.0) \times 10^{-11}$ cm$^3$ molecule$^{-1}$ s$^{-1}$ where the quoted error results from the 2σ statistical error of the weighted mean and an
additional 10% relative error to cover uncertainties derived from the experimental and evaluation procedure.

170          The rate coefficient of 3P2 has been determined previously in our laboratory ($k_{3P2} = (7.22 \pm 1.74) \times 10^{-11}$ cm$^3$
molecule$^{-1}$ s$^{-1}$) based on experiments employing a 3P2 sample, which contained about 30% mesityl oxide as major impurity
(Blanco et al., 2012). Nevertheless, both determinations agree within 20%.

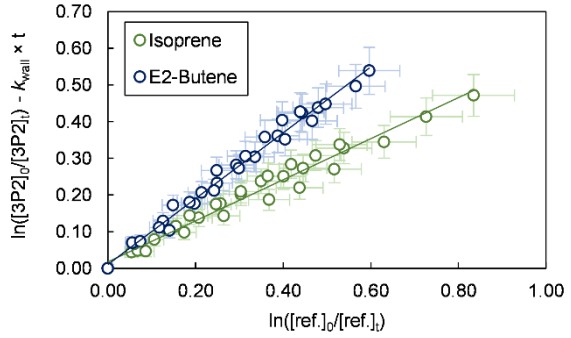

**Figure 1.** Relative-rate plots of all experiments using isoprene and E2-butene as references. The error bars consist of a systematic uncertainty
and an additional 10% relative error to cover uncertainties derived from the experimental and evaluation procedure, respectively.



**Table 1.** Results of the 3P2 + OH kinetic experiments.

| Experiment | Reference | $k_{3P2}/k_{ref.}$ | $k \times 10^{11}$ / cm$^3$ molecule$^{-1}$ s$^{-1}$ |
|---|---|---|---|
| 3P2#1[a] | Isoprene | 0.60 ± 0.03 | 6.0 ± 1.0 |
| 3P2#2[a] | E2-Butene | 0.89 ± 0.04 | 6.3 ± 1.0 |
| 3P2#3[a] | E2-Butene | 0.97 ± 0.06 | 6.9 ± 1.1 |
| 3P2#4[a] | Isoprene | 0.63 ± 0.04 | 6.3 ± 1.0 |
| 3P2#5[b] | E2-Butene | 0.89 ± 0.04 | 6.3 ± 1.0 |
| 3P2#6[b] | Isoprene | 0.56 ± 0.03 | 5.6 ± 0.9 |
| | | **Average** | **6.2 ± 1.0** |

[a] Performed in the 1080 L chamber, [b] performed in the 480 L chamber.


### 3.2 In situ generation of 2-hydroxypropanal

2-Hydroxypropanal (2HPr) is one of the expected main products in the OH radical initiated oxidation of 3P2. This α-hydroxyaldehyde, commonly known as lactaldehyde, is available commercially only as 1 M aqueous solutions, where various dimer species coexists (Takahashi et al., 1983). Therefore, in order to obtain gas-phase FTIR spectra and cross sections of

2HPr, the ozonolysis reactions of 3-buten-2-ol (3B2OL) and 3-penten-2-ol (3P2OL) were used to generate the α-hydroxyaldehyde in situ in the 1080 L chamber. Given the wall loss of 2HPr, observed in most of the 3P2 product study experiments in the 1080 L chamber (see Sect. 3.3), the 3B2OL ozonolysis experiments were optimized to ensure large product formation ratios with negligible losses.

It is well established that ozonolysis reactions proceed through a 1,3-dipolar cycloaddition forming initially a five-

membered primary ozonide (POZ) which readily decomposes through bond scission between the carbon atoms of the former C=C bond and one of the O-O bonds. Hence, assuming this to be the only reaction pathway taking place, each of the two possible decomposition channels results in the formation of a stable carbonyl species and a biradical (CI = Criegee Intermediate) as shown in Figure 2. 3-Buten-2-ol ozonolysis leads to formaldehyde as one of the primary carbonyls. When comparing the product spectra obtained here to IR spectra of 1,2-epoxybutane there is no indication for epoxide formation in

the 3B2OL ozonolysis spectra thus giving confidence in the correctness of the initial assumption. Since the sum of the yields of both HCHO and 2HPr, expected to be the other primary carbonyl (Fig. 2), should accordingly be equal to unity, then the 2HPr yield ($y_{2HPr}$) is defined as $1 - y_{HCHO}$. This allows to derive a correlation between the integrated absorption of 2HPr and its concentration. Beside HCHO and 2HPr acetaldehyde formation can be observed unambiguously in the 3B2OL ozonolysis system. The acetaldehyde yield was reproducible throughout all 3B2OL experiments. The formation can only be explained by

decomposition of the CI formed according to channel (a) (Figure 2). Our results show that acetaldehyde accounts for 36 ± 10%





of the CI decomposition in the 3B2OL ozonolysis. However, the fate of CIs is sensitive to the degree of excitation, which is not necessarily the same for different ozonolysis systems. Given that the same CI is formed in the 3P2OL ozonolysis (Figure 2) acting as potential secondary source of acetaldehyde this experimental system turned out improper for a cross section determination of 2HPr. The 3P2OL ozonolysis has therefore only been used for confirmation of the 2HPr spectral features by

comparison with the 3B2OL experimental system (see Sect. 3.3).

**Figure 2.** Formation of 2-hydroxypropanal through the ozonolysis of 3-buten-2-ol (3B2OL, red) and 3-penten-2-ol (3P2OL, blue) and respective average branching ratios.


The observed yield for the primary aldehyde HCHO ($0.38 \pm 0.06$) is well below 0.5, thus indicating a preference of the POZ decomposition towards 2HPr formation. Since the OH absorption band centred around 3550 cm$^{-1}$ (integration range: 3580–3500 cm$^{-1}$) is expected to be free from interferences of other unidentified species this band was chosen to determine a cross section for 2HPr based on the comparison of FTIR spectra of both ozonolysis systems and residuals of the 3P2 + OH

system. A plot of the integrated OH absorption versus the concentration of 2HPr, calculated based on the yields as discussed above, is shown in Figure 3. The cross section for the integrated OH absorption band was determined as $(5.0 \pm 1.5) \times 10^{-18}$ cm molecule$^{-1}$ (base e) by averaging the results of all individual experiments. The experimental results agree within 9% but given the error of the HCHO yield and the wall loss of 2HPr we prefer, however, to assign an expanded uncertainty of 30%.





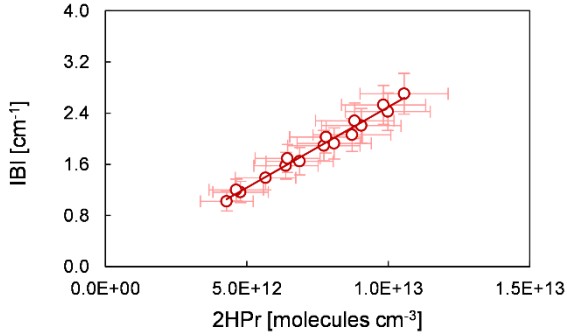


**Figure 3.** Correlation between the integrated absorption band of 2HPr in the range 3580–3500 cm$^{-1}$ and the concentration of 2HPr. The experiments were carried out at an optical path length 484.7 ± 0.8 m.

### 3.3 3-Penten-2-one + OH mechanism


Product study experiments were carried out in both simulation chambers. After subtraction of 3P2 and the species related to the methyl nitrite photolysis itself (methyl nitrite, methyl nitrate, NO, $NO_2$, HONO, $HNO_3$, and HCHO) the FTIR spectra contain absorption features that can be attributed unambiguously to acetaldehyde, methyl glyoxal and peroxyacetyl nitrate (PAN). $CO_2$ formation is observed by the absorption features in the range 2400–2250 cm$^{-1}$. Plots of the identified products (in ppmV) versus the consumed 3P2 corrected for the wall loss (in ppmV) are presented in Figure 4 for all conducted experiments.


In the case of methyl glyoxal and acetaldehyde these correlations exhibit a very high linearity. This indicates clearly their formation as primary products in the OH initiated oxidation of 3P2. Molar formation yields based on averaging the results of the regression analysis of each experiment are 0.40 ± 0.07 and 0.29 ± 0.09 for acetaldehyde and methyl glyoxal, respectively, without corrections other than the 3P2 wall loss. The errors consist of the 2σ statistical error of the mean and an additional 10% relative error to cover uncertainties derived from the evaluation procedure. The larger relative error associated with methyl glyoxal results from a larger scattering in the experimental data.


Based on the SAR-approach provided by Kwok and Atkinson (1995) the group rate constant is in the order of 10$^{-13}$ cm$^3$ molecule$^{-1}$ s$^{-1}$ for H-atom abstraction from the terminal methyl groups whereas addition to the C=C double bond accounts for about 10$^{-11}$ cm$^3$ molecule$^{-1}$ s$^{-1}$. Hence, it is very likely that the OH radical will add almost exclusively to either the α- or β-carbon atom to form the corresponding α- or β-hydroxyalkyl radical. Under atmospheric conditions these radicals will react immediately with molecular oxygen yielding the corresponding hydroxyperoxy radicals. By employing an excess of NO virtually all $RO_2$ radicals will react with NO and form mainly hydroxyalkoxy radicals (Figure 5). A fraction of the $RO_2$ + NO reaction might also produce organic nitrates ($RONO_2$). The β-RO radical could react either with oxygen to yield a 1,3-dicarbonyl species (Figure 5, pathway $\beta_2$) or dissociate to form acetaldehyde. The co-built hydroxyalkyl radical will react subsequently with oxygen yielding methyl glyoxal, in agreement with our experimental observations.







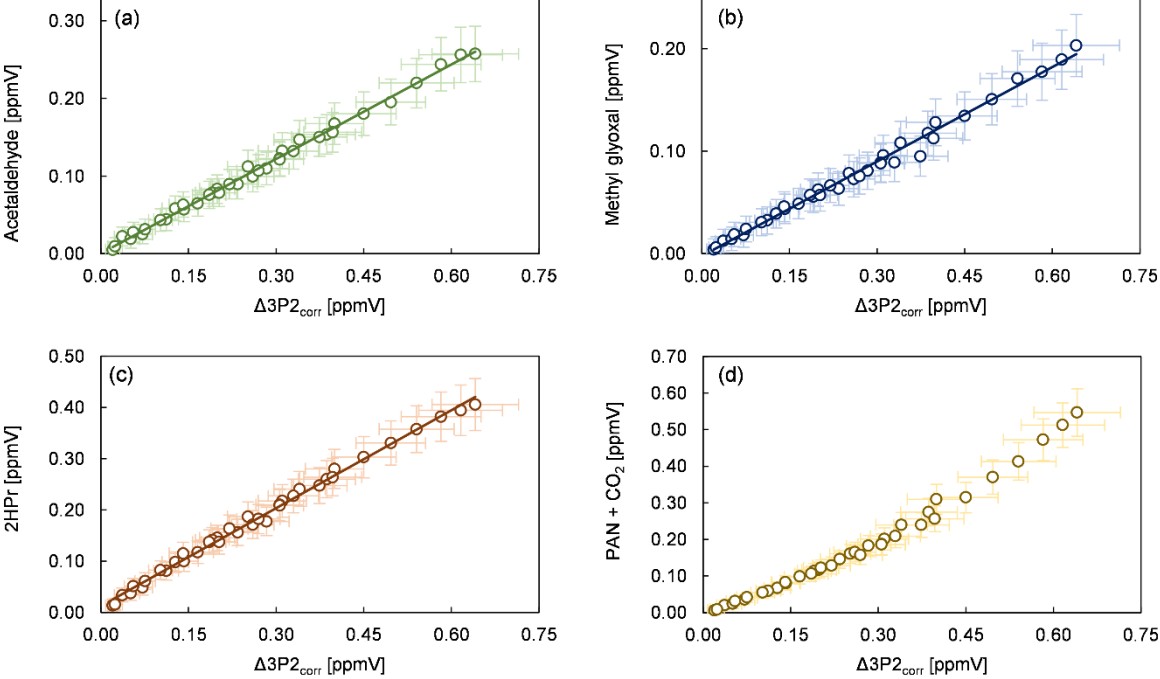

**Figure 4.** Yield plots for (a) acetaldehyde, (b) methyl glyoxal, (c) 2-hydroxypropanal, and (d) the sum of PAN and $CO_2$ for all conducted experiments corrected for the wall loss of 3P2. The error bars consist of a systematic uncertainty and a 10% relative error. The data of the 480 L chamber experiments are multiplied with a factor of 0.1 to fit within the scale of 1080 L chamber experiments.


Formation of PAN originates from $CH_3C(O)$ radicals, generated in the reaction system through the subsequent reactions with oxygen and $NO_2$. However, we recently pointed out that under the typical experimental conditions when the methyl nitrite photolysis is used as source for OH radicals, PAN formation accounts only for up to one third of the fate of acetyl radicals (Illmann et al., 2021b). The main fate of the readily formed acetylperoxy radical will be the reaction with NO

which eventually yields $CO_2$ and HCHO. Since HCHO is formed in the methyl nitrite photolysis itself the sum of PAN and $CO_2$ has been used to estimate the formation yield of $CH_3C(O)$ radicals. The correlation between (PAN + $CO_2$) and Δ3P2 is strongly linear up to a consumption of about 30% (Figure 4) and becomes precisely non-linear with higher levels of the 3P2 consumption. This indicates strongly, on the one hand, the formation of $CH_3C(O)$ radicals due to the OH reaction of 3P2. On the other hand, secondary processes like further oxidation of the first generation products increase the formation rate of acetyl

radicals at longer reaction times. The primary generation of $CH_3C(O)$ radicals results from the bond scission between $C_\alpha$ and the carbonyl carbon atom of the α-RO radical (pathway $α_1$, Figure 5). The molar formation yield of the sum PAN + $CO_2$, without corrections, is $0.63 \pm 0.14$ based on averaging the results of the regression analysis of each experiment over the linear range.





In contrast to daytime conditions within the troposphere, the photolysis of the generated acetaldehyde and methyl
glyoxal under our experimental conditions was negligible compared to the further oxidation through OH radicals. The reaction
of both aldehydes with OH was shown to proceed nearly exclusively via the abstraction of the aldehydic H atom, therefore
being a secondary source of $CH_3C(O)$ radicals with strength of 95–100% (Calvert et al., 2011).

Similarly to the β-RO, the α-RO radical could also react with oxygen according to pathway $α_3$ to form a
hydroxydicarbonyl species (Figure 5). However, based on the observed spectral features there is no evidence to support further
transformation pathways of both RO radicals other than the decomposition channels. This is in agreement with the product
studies of structurally similar α,β-unsaturated ketones conducted under conditions where RO radicals are formed solely through
the reaction of $RO_2$ + NO (Tuazon and Atkinson, 1989; Galloway et al., 2011; Praske et al., 2015; Illmann et al., 2021b). The
exothermicity of $RO_2$ + NO results in RO radicals that are chemically activated and prone to decomposition (Orlando et al.,
2003). For 1,2-hydroxyalkoxy radicals it was also shown that the reaction with $O_2$ cannot compete with the dissociation channel
(Atkinson, 2007).

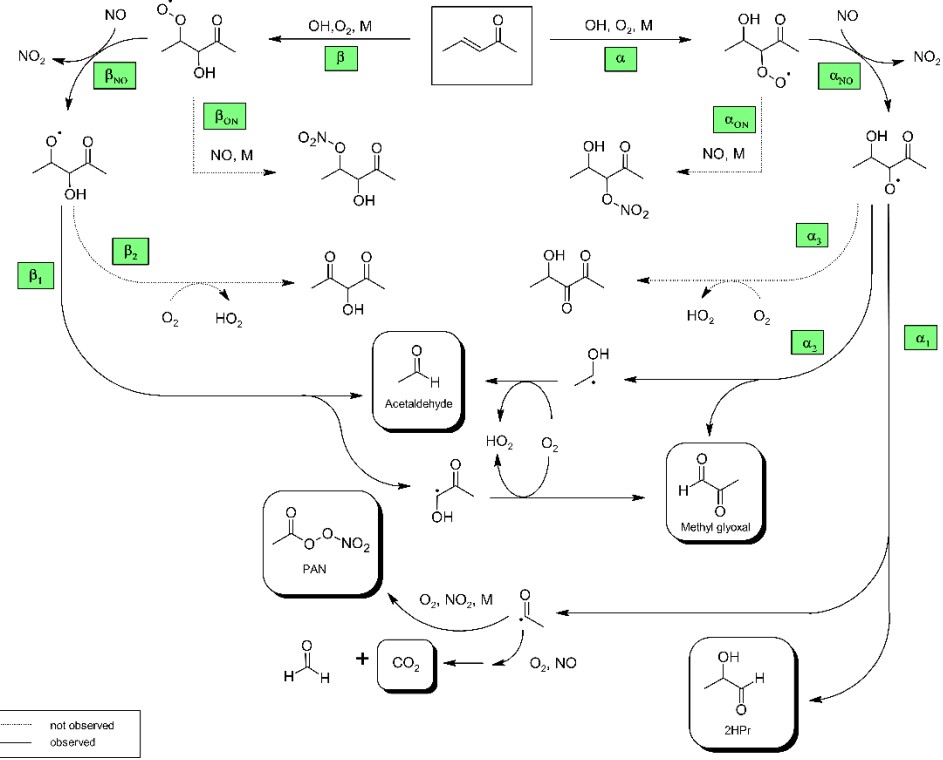

**Figure 5.** Proposed mechanism for the OH radical initiated oxidation of 3P2. The reaction pathways are named according to the position
where the oxygen adds to form the peroxy radical.






2-hydroxypropanal is a co-product of $CH_3C(O)$ radicals in the pathway $\alpha_1$ (Figure 5). Residual spectra of the 3P2 + OH system after subtraction of acetaldehyde, methyl glyoxal and PAN are shown in Figure 6 together with spectra recorded during the ozonolysis of 3B2OL and 3P2OL (s. Sect. 3.2). Absorption bands centred on 3640 cm$^{-1}$ and in the range 1120–1090 cm$^{-1}$ and centred on 3640 cm$^{-1}$, 980 cm$^{-1}$ and in the range 1120–1090 cm$^{-1}$ are present only in the residual spectra of the

3P2OL and 3B2OL ozonolysis experiments, respectively, indicating additional unidentified reaction products, which likely result from CI decomposition processes. However, other spectral ranges centred on 3550 cm$^{-1}$ (O-H stretching vibration), 1750 cm$^{-1}$ (C=O stretching vibration), 1370 cm$^{-1}$ (C-H/O-H bending vibration), 830 cm$^{-1}$ (C-H bending vibration), and main parts of the characteristic absorption pattern in the range 3040–2640 cm$^{-1}$ (C-H stretching vibration) agree within the spectra in both position and relative intensity, thus giving confidence in the identification of 2HPr.

Based on the integrated absorption cross section determined within this work an averaged molar yield of $0.59 \pm 0.25$ is derived for 2HPr from the regression analysis of all experiments without further corrections. One should note that the 2HPr yields of all experiments, performed in both chambers, agree within 12% and the major uncertainty is derived from the accuracy of the absorption cross section. Nevertheless, the 2HPr yield is similar to the molar yield of PAN + $CO_2$ which is reflected by the proposed mechanism since these species are formed accordingly in the same reaction pathway (Figure 5). On the other

hand, the yield plots of 2HPr show a small but precise curvature in each experiment. The wall loss of the $\alpha$-hydroxyaldehyde was found to be $< 10 \times 10^{-5}$ s$^{-1}$ in the 1080 L chamber and $< 4 \times 10^{-5}$ s$^{-1}$ in the 480 L chamber, respectively. This all indicates that further oxidation of the $\alpha$-hydroxyaldehyde is significant under the experimental conditions.

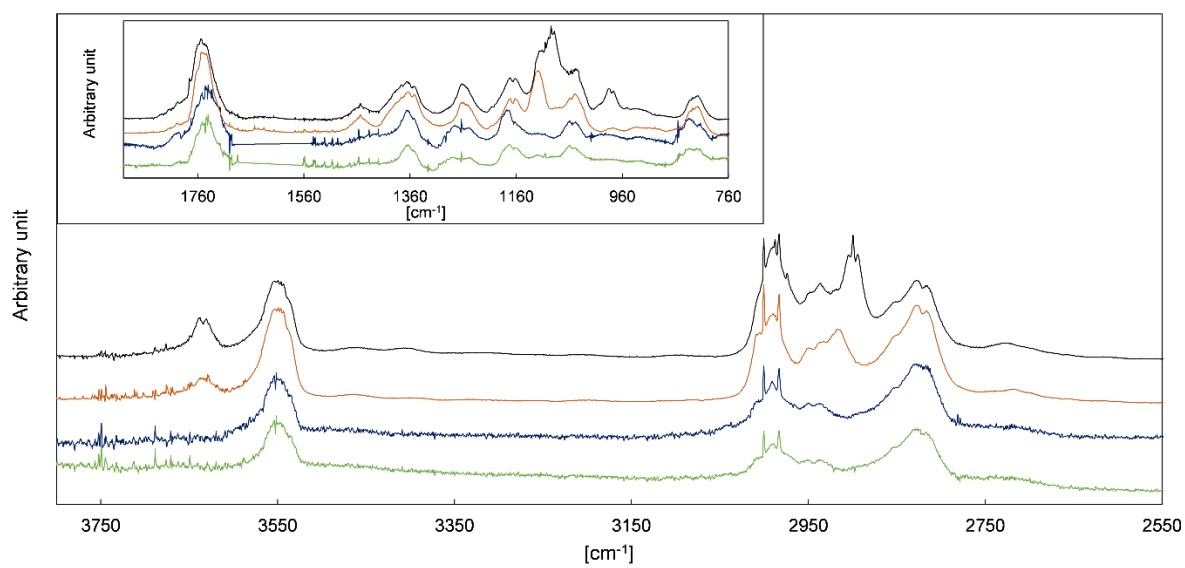

**Figure 6.** FTIR spectra of the residuals assigned to 2HPr obtained in the 1080 L chamber (green) and 480 L chamber (blue) 3P2 + OH experiments and residuals obtained from the ozonolysis of 3-penten-2-ol (brown) and 3-buten-2-ol (black).



### 3.4 2-Hydroxypropanal + OH and yields correction

Among the class of α-hydroxyaldehydes mechanistic information on the OH reaction and photolysis at atmospheric pressure
were reported merely for glycolaldehyde (Niki et al., 1987; Bacher et al., 2001; Magneron et al., 2005). Baker et al. (2004)
reported rate coefficients for the OH reaction of a series of hydroxyaldehydes synthesized in situ via the reaction of OH radicals
with precursor alcohols. The authors obtained $k = (1.7 \pm 0.2) \times 10^{-11}$ cm$^3$ molecule$^{-1}$ s$^{-1}$ for 2HPr + OH through a nonlinear
least squares analysis of the data of the 2-methyl-2,4-pentanediol oxidation (Baker et al., 2004). Under tropospheric daytime
conditions both photolysis and OH initiated oxidation are important removal processes for glycolaldehyde (Bacher et al.,
2001). However, assuming a similar behaviour for 2HPr, photolysis is expected to be negligible under the experimental
conditions of the present study, since the OH radical level is much higher while photolysis frequencies are lower than within
the troposphere. Including the above rate constant of 2HPr + OH into the model described previously (Illmann et al., 2021b)
results in a corrected average yield of $0.68 \pm 0.27$, which is about 15% higher than determined from the yield plot, without
proper corrections (Figure 4). Hence, a significant fraction of the α-hydroxyaldehyde is subject to OH radical initiated
oxidation.

Based on the SAR approach by Kwok and Atkinson (1995) and the mechanistic information reported for the
glycolaldehyde oxidation (Niki et al., 1987; Bacher et al., 2001; Magneron et al., 2005) one would expect abstraction of the
aldehydic H atom to dominate compared to abstraction of the carbon-bonded H atom of the –CH(OH)– entity for the OH
reaction of 2HPr, as presented in Figure 7. The abstraction from the terminal –CH$_3$ group and the –OH group is expected to
be negligible due to the much lower group rate constants. The hydroxypropionyl radical formed according to channel (a) will
either eliminate carbon monoxide and react with O$_2$ to form acetaldehyde or react with oxygen to form a
hydroxypropionylperoxy radical (Figure 7). The latter radical, resulting from channel (2a), may either yield
peroxyhydroxypropionyl nitrate or will be converted to the corresponding RO radical (Figure 7). This species will readily
eliminate CO$_2$ and finally form acetaldehyde as well. By analogy to the OH initiated oxidation of 3-hydroxy-2-butanone
(Aschmann et al., 2000) one would expect reaction with oxygen to predominate over decomposition for the hydroxyalkyl
radical formed following channel (b), thus leading to methyl glyoxal (Figure 7). Hence, the 2HPr + OH reaction is expected
as a secondary source of acetaldehyde and methyl glyoxal in the experimental system.

In order to investigate the 2HPr + OH reaction, methyl nitrite and NO were added for a second time, after a 3P2
consumption of about 70%, to shift the reaction system towards secondary oxidation processes. Applying the approach
presented by Baker et al. (2004) to these experiments yields a value of $(2.2 \pm 0.6) \times 10^{-11}$ cm$^3$ molecule$^{-1}$ s$^{-1}$ for the rate
coefficient of 2HPr + OH which is about 30% larger than previously reported (Baker et al., 2004). Given that both
determinations are based on the in situ generation of the α-hydroxyaldehyde this is still an excellent agreement. As shown in
Figure 8, 2HPr (green) passes through a small maximum during the second irradiation period. The mixing ratio of acetaldehyde
(purple) still increases while reaching a plateau at the end of the reaction in the case of methyl glyoxal (black). This is in
qualitative agreement with the proposed mechanism. Peroxy nitrates other than PAN, formed through 3P2 + OH, could not be





detected. Traces of the analogue peroxyhydroxyacyl nitrate resulting from glycolaldehyde oxidation have only been observed when the corresponding $RO_2$ radical was generated through the reaction of glycolaldehyde with Cl atoms in the presence of $NO_2$ (Niki et al., 1987). Magneron et al. (2005) did not detect any PAN-type species in the glycolaldehyde + OH system and concluded that this species is likely unstable and readily dissociates. Hence, abstraction of the aldehydic H atom following

channel (a) will likely result exclusively in the formation of acetaldehyde (Figure 7).

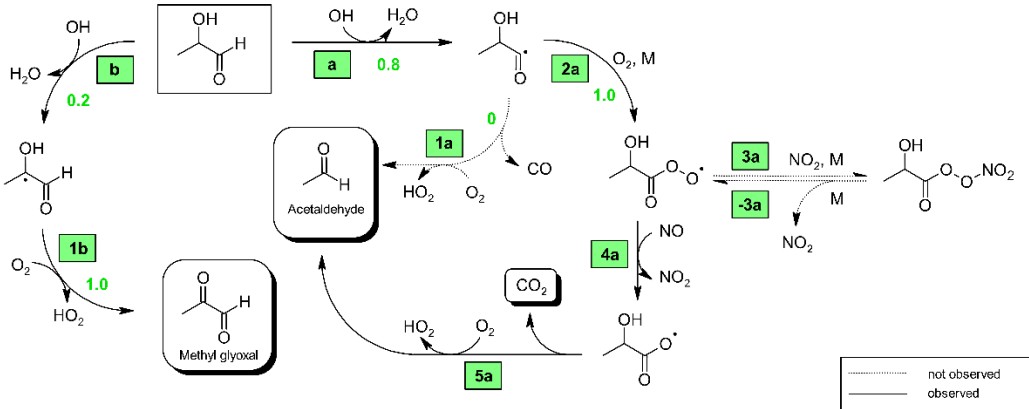

**Figure 7.** Proposed mechanism for the OH radical initiated oxidation of 2-hydroxypropanal (2HPr). Relevant branching ratios estimated within this work are denoted in green next to the reaction channel.


    Molar formation yields for acetaldehyde and methyl glyoxal, derived from 3P2 + OH, and the branching ratio $k_b/[k_a + k_b]$, resulting in methyl glyoxal through 2HPr + OH, were included as parameters in the model and varied until the experimental time profiles are reproduced by the simulation. Since the OH initiated oxidation of 2HPr is expected to proceed solely through (a) and (b) the branching ratio $k_a/[k_a + k_b]$ is given as $1 - k_b/[k_a + k_b]$ (Figure 7). The secondary oxidation of

both acetaldehyde and methyl glyoxal is almost negligible immediately after switching on the lamps for the first time. On the other hand, as stated above, secondary oxidation processes dominate in the end of the second irradiation period. Accordingly, only a small variation in the parameters' values allows reproducing the time profile of the entire experiment and the branching ratios $k_a/[k_a + k_b]$ and $k_b/[k_a + k_b]$ were found to be $0.79 \pm 0.05$ and $0.21 \pm 0.05$, respectively. This is in excellent agreement with SAR predictions (Kwok and Atkinson, 1995) which estimate 0.8 and 0.2 for the branching ratios, respectively, as well as

former results on the OH reaction of glycolaldehyde at atmospheric pressure (Niki et al., 1987; Bacher et al., 2001; Magneron et al., 2005).

    Based on these results the corrected yields for acetaldehyde and methyl glyoxal from the 3P2 + OH reaction are $0.39 \pm 0.07$ and $0.32 \pm 0.08$, respectively. Hence, while larger molar yields were observed for acetaldehyde than for methyl glyoxal without proper corrections the model predicts both first-generation yields to be the same within the accuracy errors, which

proves their formation according to the same reaction channel.



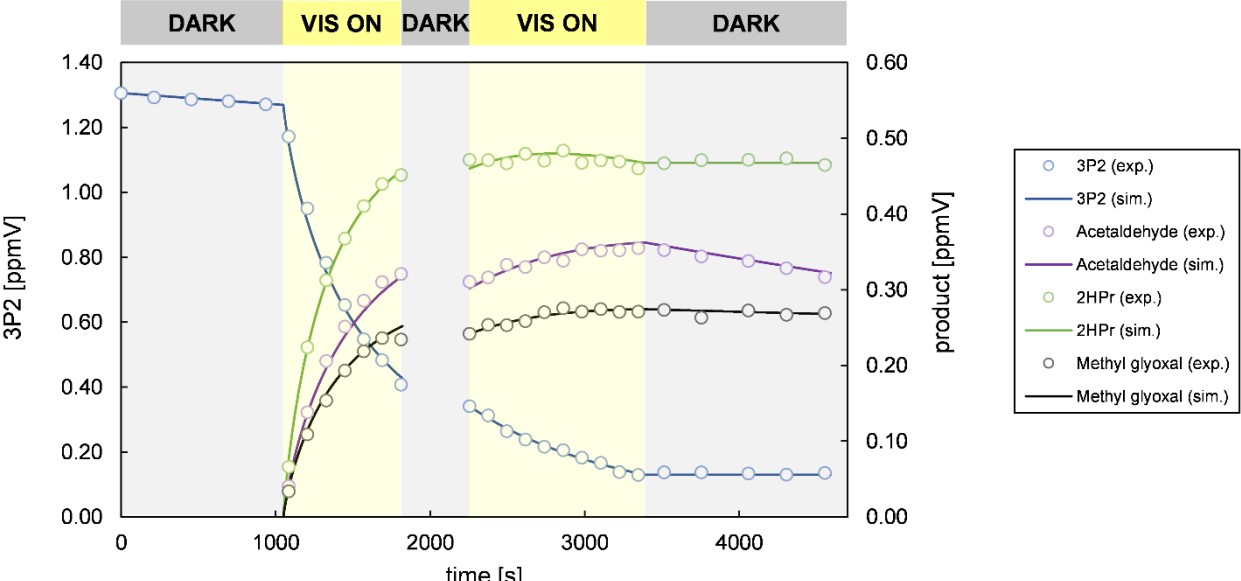

**Figure 8.** Time profile for a 3P2 + OH experiment, performed in the 1080 L chamber, with a second addition of methyl nitrite and NO during the second dark phase of the experiment. The experimental set-up did not allow the quantification of $CO_2$.


By considering the formation of $CH_3C(O)$ radicals from the oxidation of 3P2, acetaldehyde and methyl glyoxal the model underestimates PAN + $CO_2$ at longer reaction times as depicted by the "PAN+CO2_v1" simulation in Figure 9. This can be partly explained by an additional source of $CO_2$ in the experimental system, since it is also a co-product of acetaldehyde via channel (5a) in the 2HPr + OH reaction (Figure 7). Given that abstraction of the aldehydic H atom of 2HPr is expectedly

leading solely to acetaldehyde, the yield of $CO_2$ from 2HPr oxidation depends only on the ratio between decomposition of the hydroxypropionyl radical and its reaction with oxygen (Figure 7). Méreau et al. (2001) concluded that decomposition cannot compete with the $O_2$ reaction in the case of the structurally similar hydroxyacetyl radical, based on *ab initio* calculations. Besides, Niki et al. (1987) observed $CO_2$ instead of CO formation in the glycolaldehyde oxidation when secondary oxidation processes were minimized in the experimental system. These findings together with the discrepancy of simulated and

experimental time profile for PAN + $CO_2$ at long irradiation times suggest that decomposition of the hydroxypropionyl radical is negligible and $k_{2a}/[k_{1a} + k_{2a}] = 1$ (see Figure 7). Including the additional $CO_2$ source in the model improves significantly the consistency between the simulated and experimental PAN + $CO_2$ profile at long irradiation times. As shown in Figure 9 for a 480 L chamber experiment the entire time profile of PAN + $CO_2$ is reproduced when the additional source of carbon dioxide is included into the model. One should note that in this regard the time profile does no longer represent merely the formation

of acetyl radicals. However, given that both the simulation with and without the additional $CO_2$ source are indistinguishable



in the first part of the irradiation period (Figure 9) it is still possible to derive the corrected average yield for PAN + $CO_2$ (0.56 ± 0.14) representing the yield of $CH_3C(O)$ radicals.

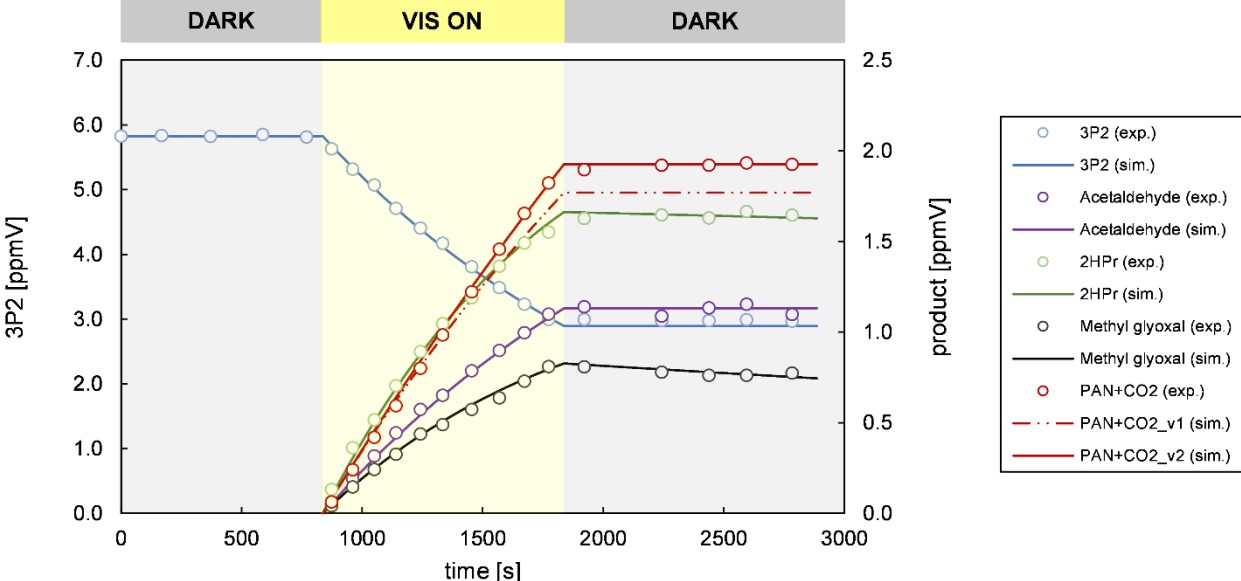

**Figure 9.** Temporal behaviour of 3P2, 2HPr, acetaldehyde, methyl glyoxal and PAN + $CO_2$ in an experiment performed in the 480 L chamber. The "PAN+CO2_v1" simulation considers only the formation of PAN and $CO_2$ due to $CH_3C(O)$ radicals formed in the reaction system. The "PAN+CO2_v2" simulation includes also the additional $CO_2$ source from 2HPr + OH.

The lowering of the PAN + $CO_2$ yield due to the correction is consistent with the presence of secondary processes since both acetaldehyde and methyl glyoxal further oxidation contributes to the $CH_3C(O)$ radical formation in the experimental system. Besides, as for acetyldehyde and methyl glyoxal, the yields for 2HPr and PAN + $CO_2$ are the same within the assigned accuracy thus proving their formation in the same reaction channel. As carbon dioxide formation might be easily affected from processes on the chamber walls and the corrected yield for PAN + $CO_2$, therefore, be still regarded as upper limit. A build-up of $CO_2$ from the walls might become relevant at longer irradiation times and this supposedly explain the remaining small discrepancies at irradiation times > 10 min in some experiments. However, the reproducibility of the yields without correction is essentially the same as for 2HPr for experiments performed in both chambers. Besides, separate control experiments in which synthetic air was irradiated with the same set of lamps did not show significant $CO_2$ production. Therefore, the influence of off-gasing processes on its temporal behaviour is probably negligible in the beginning of the experiments, when the formation of the products in the target reaction dominates over secondary chemistry. An overestimation of the $CH_3C(O)$ radical yield is thus unlikely. Uncorrected and corrected molar yields, namely first-generation yields, of all quantified products are summarised in Table 2.





Combining the yields of the 3P2 oxidation products leads to a carbon balance close to unity (0.98 ± 0.18). The branching ratios for the pathways $\alpha_{ON}$ and $\beta_{ON}$ (Fig. 5) forming $RONO_2$ species are expectedly very minor channels. This is in agreement with previous work in which $RONO_2$ species from the OH oxidation of α,β-unsaturated ketones were indicated only
in our experimental set-up resulting from tertiary $RO_2$ radicals (Illmann et al., 2021b). Besides, Praske et al. (2015) reported a low overall $RONO_2$ yield of 0.040 ± 0.006 for MVK oxidation.

**Table 2.** Uncorrected yields and first-generation yields (yields corrected for secondary processes) of species identified in the 3P2 + OH reaction.

|  | 2-Hydroxypropanal | PAN + $CO_2$ | Methyl glyoxal | Acetaldehyde |
|---|---|---|---|---|
| uncorrected | 0.59 ± 0.25 | 0.63 ± 0.14 | 0.29 ± 0.09 | 0.40 ± 0.07 |
| corrected | 0.68 ± 0.27 | 0.56 ± 0.14 | 0.32 ± 0.08 | 0.39 ± 0.07 |


## 4. Atmospheric implication and conclusion

The atmospheric lifetime with respect to OH radicals, defined as $1/(k_{3P2} \times [OH])$, is about 4.5 h when assuming a global average OH radical concentration of $1.0 \times 10^6$ cm$^{-3}$ within the troposphere (Bloss et al., 2005). Calvert et al. (2011) recommended,
however, an OH radical level of $2.5 \times 10^6$ cm$^{-3}$ for the calculation of atmospheric lifetimes for short-lived species which leads consequently to an even shorter lifetime. Since our experiments indicate no measurable photolysis, the OH reaction is the dominant degradation process during daytime. 3P2 is thus oxidized close to the emission/formation source. A larger influence on atmospheric processes is indicated by the primary formation of $CH_3C(O)$ radicals, which account for 22 ± 6% of the 3P2 oxidation. As acetyl radicals yield ultimately PAN, depending on the $NO_2/NO$ ratio, 3P2 exhibits a huge potential of forming
$NO_x$ reservoir species. The gas-phase oxidation of the first-generation product 2HPr by OH radicals would generate mainly acetaldehyde (~80%) hence increasing the potential of forming $NO_x$ reservoirs. However, by comparison with glycolaldehyde photolysis of 2HPr may also be competitive. On the other hand, since 2HPr is highly soluble in water, uptake into the aqueous phase (aerosols) may also be an important loss process. Hydration would increase significantly the lifetime towards photolysis and lead potentially to the formation of organic acids, as discussed previously for glycolaldehyde (Calvert et al., 2011).
Given both the short lifetime and the mechanism of the OH initiated oxidation, 3P2 is an example of species those chemistry explains the rapid PAN formation in young biomass burning plumes, as found previously in field observations (Alvarado et al., 2010). Acetaldehyde and methyl glyoxal form as well acetyl radicals in their further gas-phase oxidation (Calvert et al., 2011). Besides, methyl glyoxal is also known as a source of secondary organic aerosol (Fu et al., 2008). Box-models yielded high average OH radical concentrations of about $7.5 \times 10^6$ cm$^{-3}$ within young biomass burning plumes (Müller
et al., 2016). Based on that, once emitted from biomass burning, more than 50% of the 3P2 carbon are converted into $CH_3C(O)$



radicals in less than 3 h. Assuming gas-phase products only in the further oxidation pathways possibly up to 2 PAN molecules are formed per 3P2 molecule consumed. The α,β-unsaturated ketone fits also well into the characteristics of unknown VOCs as deployed by Alvarado et al. (2015) to elucidate the evolution of $O_3$ and secondary organic aerosol in a plume of a prescribed fire in California since (a) the OH rate coefficient of 3P2 is in the order of $10^{-11}$ cm$^3$ molecule$^{-1}$ s$^{-1}$, (b) the $RO_2 + NO$ reactions

of 3P2 derived peroxy radicals result exclusively in fragmentation of the molecule, and (c) the mechanism proposed in this study predicts a high $HO_2$ regeneration level. Therefore, single-component studies as the present one contribute to a better understanding of the complex biomass burning plume chemistry.

*Data availability.* Data can be provided upon request from the corresponding author.


*Author contribution.* NI conducted the experiments and processed the data. NI prepared the manuscript with contributions from all co-authors.

*Competing interests.* The authors declare that they have no competing interests.


*Acknowledgements.* The authors gratefully acknowledge funding from the EU Horizon 2020 research and innovation programme through the EUROCHAMP-2020 Infrastructure Activity (grant agreement no. 730997) and the Deutsche Forschungsgemeinschaft (DFG) through the grant agreement WI 958/18-1.

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
