# Peer review of "Biomass burning plume chemistry: OH radical initiated oxidation of 3-penten-2-one and its main oxidation product 2-hydroxypropanal"

_Atmospheric Chemistry and Physics, 2021_

## Author Comment (AC1)

**Authors' responses to comments from referee 1 on:** Illmann et al., Atmos. Chem. Phys. Discuss., https://doi.org/10.5194/acp-2021-575

*We thank the referee for the valuable comments on this work. The original comments are shown in black and our responses are marked in blue. Changes made in the text are marked in red.*

Comment 1: In this work, the authors provide a kinetic and mechanistic study of the OH-initiated oxidation of 3-penten-2-one (3P2), and its by-product 2-hydroxy-propanal (2HPr), chemistry that is of potential importance in biomass burning plumes. Major outcomes of the work include a first-time determination of the products of the OH-initiated oxidation of the two species, and a determination of the rate coefficients for their OH reaction (each of which was the subject of one previous study).

It is my opinion that this is likely to be a publishable body of work, although I think there are a few quantitative issues that need to be resolved before publication is recommended.

Response: We thank the referee for the evaluation and the very useful suggestions.

Comment 2: My major comment has to do with the modeling work that is done. I think much more needs to be said about the details of this modeling (even though this is outlined in a recent paper by the same authors), to lend confidence in the key parameters that are derived from it (e.g., k(OH+2HPr), branching ratios for this reaction, 'corrected' yields of various products). First, the details of what is included in the model, how it is run, what parameters are varied, etc., needs to be included. Further, the sensitivity of the model output to the varied parameters needs to be shown (presumably graphically). As examples, how sensitive is the model to the OH/2HPr rate coefficient, to the branching ratios to the two channels in this reaction, etc. What does an 80:20 branching in the OH/2HPr reaction look like compared to 70:30 or 90:10? Etc…

Response: The referee's comment contains quite a lot of aspects. In the following we thus subdivided the comment and we will try to answer point-by-point to the different issues that were addressed.

My major comment has to do with the modeling work that is done. I think much more needs to be said about the details of this modeling (even though this is outlined in a recent paper by the same authors), to lend confidence in the key parameters that are derived from it (e.g., k(OH+2HPr), branching ratios for this reaction, 'corrected' yields of various products). First, the details of what is included in the model, how it is run, what parameters are varied, etc., needs to be included.

Response: On the one hand, we understand from this comment that the referee is missing some details of the modelling work. The approach we use can be written down in calculation programs like Microsoft Excel. It does not aim to fully draw a detailed mechanism, with all radical reactions (like MCM), and is just used to describe the temporal behaviour of the main species in chamber experiments with a defined radical source like methyl nitrite photolysis. In doing so, one can obtain the product yields (branching ratios) of the target reaction. Based on simplified reaction sequences (for instance: A + OH → B + C, A + wall, B + OH → C, C + OH → …) differential equations are constructed for each experimentally quantified species. Input parameters, as described in our recent paper, are the initial concentration of A (which corresponds to 3P2), the rate coefficient for each reaction and the OH-concentration. The whole calculation is based on the Euler method (= Euler-Cauchy approach) which is the most basic numerical procedure to solve differential equations. The

time intervals (= step size), in which the differential equations are solved, are typically in the order of < 0.1 s. The branching ratios (corresponding to the first-generation yield) of the simplified reaction scheme are included as parameters and varied until the simulated profile matches the experimental data (but no rate coefficient is determined from the model). To avoid misunderstandings we modified the brief model description (page 4, lines 120–124) as follows:

"In order to obtain first-generation yields  (= yields without impact of secondary reactions in the experimental set-up)  the temporal behaviour of all quantified species was simulated following the approach previously outlined in the recent literature (Illmann et al., 2021b). Here, the differential equations are constructed based on the simplified reaction sequence of each species and solved by the Euler method using calculation programs like Microsoft Excel. Input parameters are the rate coefficients of each of the sequence's reaction, the initial concentration of 3P2 and a time-dependent OH-concentration calculated based on the 3P2 consumption. The molar formation yields for products of the target reaction are included as variable parameters to be modified  until the simulated temporal behaviour of each species matches the experimental data. The details are provided within Sect. 3.4."

For clarity we added as well an additional table listing the simplified reaction sequence needed to describe the reaction system. This includes the reaction scheme as well as the input (kinetic) data and the parameters (first-generation yields) derived from the modelling. Given that our approach does not fully describe the mechanism, the obtained branching ratios for the reactions listed in the table 2 represent the first-generation yields and not exact branching ratios in the case of 3P2 + OH. For example, the average first-generation yield of acetaldehyde obtained through modelling is 0.38, which corresponds to the branching ratio R4b/R4 according to the simplified reaction scheme. However, given that according to our proposed mechanism acetaldehyde can be formed from addition to both $C_\alpha$ and $C_\beta$, one can just conclude that 38% acetaldehyde are formed from 3P2 + OH. Since it is not possible to decipher the branching ratios of the different RO radical decomposition channels, no further statement is possible on the exact branching ratios. By contrast, in the case of 2HPr, the obtained first-generation yields correspond to the branching ratios for the different abstraction channels.

**Table 2.** Simplified reaction sequence used for the modelling of the temporal behaviour of experimentally quantified species in order to obtain first-generation yields for the respective reactions (3P2 + OH, 2HPr + OH).

| Reaction | | Branching ratio | Rate coefficient | |
|---|---|---|---|---|
| R4 | $CH_3CH=CHC(O)CH_3 + OH \rightarrow$ products | | $6.2 \times 10^{-11}$ cm$^3$ molecule$^{-1}$ s$^{-1}$ | [a, b] |
| R4a | $\rightarrow CH_3C(OH)C(O)H$ | 0.68 [c] | | |
| R4b | $\rightarrow CH_3C(O)H$ | 0.39 [c] | | |
| R4c | $\rightarrow CH_3C(O)C(O)H$ | 0.32 [c] | | |
| R4d1 | $\rightarrow CH_3C(O)OONO_2$ | [d] | | |
| R4d2 | $\rightarrow CO_2 + HCHO$ | [d] | | |
| R5 | $CH_3CH=CHC(O)CH_3 +$ wall $\rightarrow$ | | $\leq 1.0 \times 10^{-4}$ s$^{-1}$ | [a, e] |
| R6 | $CH_3C(O)H + OH \rightarrow$ products | | $1.5 \times 10^{-11}$ cm$^3$ molecule$^{-1}$ s$^{-1}$ | [a, f] |
| R6a | $\rightarrow CH_3C(O)OONO_2$ | [a, g] | | |

| R6b | $\rightarrow CO_2 + HCHO$ | | | [a, g] |
|------|---------------------------|---|------------------------------------------------|--------|
| R7 | $CH_3C(O)H + wall \rightarrow$ | | $\leq 1.0 \times 10^{-4}$ s$^{-1}$ | [a, e] |
| R8 | $CH_3C(O)C(O)H + OH \rightarrow$ products | | $1.3 \times 10^{-11}$ cm$^3$ molecule$^{-1}$ s$^1$ | [a, h] |
| R8a | $\rightarrow CO + CH_3C(O)OONO_2$ | [i] | | |
| R8b | $\rightarrow CO + CO_2 + HCHO$ | [i] | | |
| R9 | $CH_3C(O)C(O)H + wall \rightarrow$ | | $\leq 1.0 \times 10^{-4}$ s$^{-1}$ | [a, e] |
| R10 | $CH_3C(OH)C(O)H + OH \rightarrow$ products | | $1.7 \times 10^{-11}$ cm$^3$ molecule$^{-1}$ s$^1$ | [a, j] |
| R10a | $\rightarrow CO_2 + CH_3C(O)H$ | 0.79 [c] | | |
| R10b | $\rightarrow CH_3C(O)C(O)H$ | 0.21 [c] | | |
| R11 | $CH_3C(OH)C(O)H + wall \rightarrow$ | | $\leq 1.0 \times 10^{-4}$ s$^{-1}$ | [a, e] |

[a] Input parameter; [b] rate coefficient determined within this work; [c] average branching ratio obtained through modelling; [d] average branching ratio (R4d1+R4d2)/R4 = 0.58 obtained through modelling; [e] determined in each individual experiment; [f] rate coefficient from Atkinson et al. (2006); [g] branching ratio (R6a+R6)/R6 = 0.95 from Atkinson et al. (2006); [h] rate coefficient from Atkinson et al. (2006); [i] branching ratio (R8a+R8b)/R8 = 1.0 from Atkinson et al. (2006); [j] rate coefficient from Baker et al. (2004).

On the other hand we are not sure how to interpret that "much more needs to be said about the details of this modelling (even though this is outlined in a recent paper by the same authors)". Are more details needed here than previously outlined in our recent paper? If not, in our understanding, it is not necessary to repeat the fundamentals of our approach (and a brief summary was already given in the experimental part) and we therefore think that the additional table is sufficient for traceability.

"what parameters are varied": This is basically said in the experimental section. However, for clarity this is included in the additional table and the description in the experimental section has been modified as shown above.

Further, the sensitivity of the model output to the varied parameters needs to be shown (presumably graphically) [see Response (b)]. As examples, how sensitive is the model to the OH/2HPr rate coefficient [see Response (a)], to the branching ratios to the two channels in this reaction, etc [see Response (b)]. What does an 80:20 branching in the OH/2HPr reaction look like compared to 70:30 or 90:10? Etc… [see Response (b)]

Response (a): The model was not used to determine $k$(2HPr+OH) since it would not be possible to obtain both its first-generation yield and its rate coefficient at the same time. As already written in the manuscript, the rate coefficient was estimated based on the method used previously by Baker et al. (2004). In this study the rate coefficients of in situ generated hydroxycarbonyls were obtained by a non-linear regression analysis of a plot of the formed hydroxycarbonyl vs. ln([precursor]0/[precursor]t). For this procedure, it is not necessary to include the first-generation yield of the in situ generated species since the plots profile depends only on the rate coefficient ratio $k$(hydroxycarbonyl + OH)/$k$(precursor + OH). This method is applicable only if the target species has no or at least no significant wall loss. Hence, only a very limited number of experiments allowed us to apply this approach and we therefore prefer to consider the value obtained here as an "estimated rate coefficient", as written in the abstract, and to assign an expanded uncertainty. For the modelling we used the reference value given by Baker et al. (2004). We understand, however, that how we derived $k$(2HPr+OH) was not clearly written and we would therefore include an additional figure showing the non-linear plot following the procedure outlined by Baker et al. (2004):

[Figure]

Figure 8. Non-linear plot for the formation of 2HPr from the 3P2 + OH reaction according to Baker et al. (2004) used to estimate the rate coefficient of 2HPr + OH.

We further modified the corresponding paragraph (page 13, lines 329–332) as follows:

"Applying the approach presented by Baker et al. (2004) to these experiments allows to estimate  a value of $(2.2 \pm 0.6) \times 10^{-11}$ cm$^3$ molecule$^{-1}$ s$^{-1}$ for the rate coefficient of 2HPr + OH . The non-linear plot, drawn according to the previously mentioned approach, is presented in Fig. 8. Our estimation is about 30% larger than previously reported (Baker et al., 2004). Taking into account that both determinations are based on the in situ generation of the α-hydroxyaldehyde this is still an excellent agreement."

Response (b): We thank the referee for this very valuable suggestion! Our conclusions, the given branching ratios as well as their assigned uncertainties are based on a careful sensitivity analysis. We agree, however that it might very valuable to show this graphically. We therefore updated Fig. 9 (Fig. 8 in the originally submitted manuscript) and included the temporal behaviour of acetaldehyde and methyl glyoxal, assuming different branching ratios for the 2HPr + OH reaction. We also included explicitly the sensitivity analysis in the corresponding paragraph (page 14, lines 346–360):

"Molar formation yields for acetaldehyde and methyl glyoxal, derived from 3P2 + OH, and the branching ratio $k_b/[k_a + k_b]$, resulting in methyl glyoxal through 2HPr + OH, were included as parameters in the simplified model (Illmann et al., 2021b) and varied until the experimental time profiles are reproduced by the simulation. Since the OH initiated oxidation of 2HPr is expected to proceed solely through (a) and (b) the branching ratio  is given as $1 - k_b/[k_a + k_b]$ (Fig. 7). Table 2 shows the simplified reaction sequences and the rate coefficients needed to describe the reaction system. These sequences do not follow the nomenclature used in the proposed mechanisms (Figs. 5 and 7) since the simplified model does not differentiate if a product is formed directly from a parent compound through more than one pathway.

Figure 9 summarises an analysis of the model sensitivity on the branching ratios $k_a$ and $k_b$. As can be seen in panel (a), a variation from $k_a = 1.00$ and $k_b = 0.00$ (sim1) to $k_a = 0.50$ and $k_b = 0.50$ (sim3) exhibits no measurable influence on the temporal profile in the beginning of the first irradiation period, corresponding to a 3P2 consumption of < 30%. Hence, The secondary oxidation of both acetaldehyde and methyl glyoxal was negligible immediately after switching on the lamps for the first time. This allows to derive values for the first-generation yields of acetaldehyde and methyl glyoxal from 3P2 + OH in these experiments. The branching ratios ($k_a$ and $k_b$) chosen for sim1 to sim3, depicted in panel (a), completely fail in reproducing the profile for both reaction products for the entire duration of the experiment.  Panel (b) in Fig. 9

shows the optimum range for the branching ratios that allows a simultaneously fit of the experimental time profiles for both acetaldehyde and methyl glyoxal. Accordingly,  the branching ratios $k_a/[k_a + k_b]$ and $k_b/[k_a + k_b]$ were found to be 0.79 ± 0.05 and 0.21 ± 0.05, respectively. This is in excellent agreement with SAR predictions (Kwok and Atkinson, 1995) which estimate 0.8 and 0.2 for the branching ratios, respectively, as well as former results on the OH reaction of glycolaldehyde at atmospheric pressure (Niki et al., 1987; Bacher et al., 2001; Magneron et al., 2005).

Based on these results, the temporal profiles of acetaldehyde and methyl glyoxal are well-reproduced for all conducted experiments. Their corrected yields  in the 3P2 + OH reaction are 0.39 ± 0.07 and 0.32 ± 0.08, respectively. Hence, while larger molar yields were observed for acetaldehyde than for methyl glyoxal without proper corrections, the model predicts both first-generation yields to be the same within the accuracy errors, which  indicates their formation according to the same reaction channel. The branching ratios of the simplified reaction scheme, obtained through modelling, are given in Tab. 2."

[Figure]

**Figure 9.** Experimental and simulated time profiles for a 3P2 + OH experiment, performed in the 1080 L chamber, with a  supplementary addition of methyl nitrite and NO during the second dark phase of the experiment. The experimental set-up did not allow the quantification of $CO_2$. The parameters $k_a$ and $k_b$ used in the simulation runs are: 1.00 and 0.00 (sim1), 0.90 and 0.10 (sim2), 0.50 and 0.50 (sim3), 0.84 and 0.16 (sim4), 0.79 and 0.21 (sim5), and 0.74 and 0.26 (sim6).

More minor remarks:

Comment 3: How is the 3P2 introduced into the chamber and how is its initial concentration determined? In particular, if this calibration is volumetric or similar, how is the 85% purity of the purchased sample dealt with?

Response: The 3P2 is introduced into the chamber by the injection via a Hamilton™-syringe into a heated stream of bath gas. However, this method was employed for preparing an experiment, and not used for the determination of the initial concentration. The concentrations are obtained by subtracting a calibrated

reference spectrum of 3P2 where the 3P2 cross sections were determined previously by injection of various volumes of diluted solutions (3P2 in dichloromethane) into the 480 L chamber. These highly diluted solutions are prepared from weighted 3P2 samples considering the 85w% purity.

Comment 4: Page 6 – Did the authors determine a wall loss for the reference species used in the k(OH) determinations? If so, were they negligible? What fraction of the 3P2 loss is due to the walls compared to the loss due to OH reaction in a typical experiment?

Response: The reference compounds did not show any wall loss. The ratio between the first-order loss due to OH and the wall loss was typically > 8. For clarity, we added:

"The first-order wall loss of 3P2 was < 1 ×10$^{-5}$ s$^{-1}$ in all 480 L chamber experiments and in the range of (5–10) ×10$^{-5}$ s$^{-1}$ in the 1080 L chamber experiments, respectively. Typically, the consumption through OH radicals was about one order of magnitude faster than the wall loss. The reference compounds did not show any wall loss. "

Comment 5: Page 7 – The formation of acetaldehyde from the large Criegee radical formed in the 3B2OL / O3 system is a bit surprising to me. A standard pathway for decomposition might be to form OH and methylglyoxal, rather than acetaldehyde? Can the authors provide any further ideas or speculation on its formation mechanism? (Any chance of impurities existing in the alcohol?).

Response: We were also quite surprised about the formation of acetaldehyde since we also expected rather methyl glyoxal formation through the larger CI after elimination of OH and reaction with O$_2$. Traces of methyl glyoxal might be present in the system. Due to the superposition of absorption features we can, however, neither confirm nor rule out its formation.

If acetaldehyde was formed through an impurity of the 3B2OL sample this would indicate an unidentified unsaturated species since due to the high amounts of CO, used as OH scavenger, the acetaldehyde formation must originate from an O$_3$ reaction. The purity of the 3B2OL sample is 97%. Given that the experimentally observed formation of acetaldehyde accounts for about 14% of the 3B2OL consumption this is quite unlikely resulting from an impurity. Any further speculation on the mechanism are beyond the scope of this study.

Minor corrections, suggestions, etc.:

Comment 6: I don't know what an Euler-Cauchy model is. Please explain briefly.

Response: We believe that this is a misunderstanding. We used the term "Euler-Cauchy model" just to name the approach we used. The Euler method is a basic numerical procedure to solve differential equations. However, we noticed that while in German the method is referred to "Euler-Cauchy", the English term seems to be "Euler method". We exchanged the term in the whole manuscript and modified the abstract as follows:

"Employing  a simple modelling tool to describe the temporal behaviour of the experiments, the further oxidation of 2HPr was shown …"

Comment 7: Page 2, line 33, I suggest "Unsaturated ketones are of increasing interest as …"

Response: Corrected.

Comment 8: Page 2, line 43, I suggest "is believed to be a …"

Response: Corrected.

Comment 9: Page 2, line 48, I suggest "Some field measurements of BB plumes…"

Response: Corrected.

Comment 10: Page 2, line 59, I suggest changing "Besides" to "In particular" (or something similar).

Response: We changed into "In particular".

Comment 11: Page 2, line 59, Change "expand" to "expanding".

Response: Corrected.

Comment 12: Page 6, line 167:  Maybe "all of the experimental results" instead of "the whole…"

Response: We changed into "Table 1 summarises the results obtained from all conducted kinetic experiments".

Comment 13: Page 7, line 184: 'coexist' instead of 'coexists'.

Response: Corrected.

Comment 14: Page 12, line 290: Specify that this is from the OH + 3P2 reaction.

Response: We modified as follows:

"… is derived for 2HPr in the 3P2 + OH reaction, from the regression analysis of all experiments …"

Comment 15: Page 14, line 360, also Page 16, line 392:  The equivalent yields of the two products does not prove that they come from the same pathway.  I suggest using different wording here. 'suggests' or 'consistent with'?

Response: Pages 14, line 360 has been modified as written above (Response (b) to Comment 2). We replace "proving" with "indicating" in line 392 on page 16.

Comment 16: Page 17, top:  Nitrates are quite strong absorbers in the IR.  Was there any sign of their formation? Can any limit on their yield be obtained, using nitrate IR cross sections (which are reasonably transferable) from other works?

Response: We did not find any clear indication for formation of organic nitrates. The shape of the residual spectra made any attempt to estimate an upper limit for their formation highly speculative. Therefore, we prefer to give no values.

Comment 17: Page 17, line 413:  Specify that it is 3P2 that you are referring to.

Response: We modified as follows:

"The atmospheric lifetime of 3P2 with respect to OH radicals …"

Comment 18: Page 17, line 419:  While I agree that 3P2 (and by-products) are a source of PAN, the overall contribution to NOx reservoirs will depend on the concentrations of 3P2 relative to other PAN sources.  Perhaps 'soften' the language here?

Response: The paragraph refers only to the potential of 3P2 to form $NO_x$ reservoir species and not to the contribution of this reaction to the overall $NO_x$ reservoir sources.

---

## Author Comment (AC2)

**Authors' responses to comments from Anonymous Reviewer #2 on:** Illmann et al., Atmos. Chem. Phys. Discuss., https://doi.org/10.5194/acp-2021-575

*The original comments are shown in black and our responses are marked in blue. Changes made in the text are marked in red.*

Comment 1a: This paper presents the results of a kinetic and product study of the reactions of 3-penten-2-one (3P2) with OH, which relevant to models of biomass burning plume chemistry. The results gave reasonably well characterized rate constants and product yield data, and should eventually be published. They also investigate the kinetics and mechanisms of its oxidation product 2-hydroxy propanal (2HPr) by modeling irradiations of reaction mixtures after it is formed in the experiments with 3P2. The results were used primarily as a means to correct the yield data in the 3P2 experiments for secondary reactions, but also provide a useful addition to the literature, though the results for 2HPr are more uncertain because they are obtained from modeling a complex system.

Response: We thank the referee for the comments.

Comment 1b: The use of modeling with a simplified mechanism to correct the yield data presents some uncertainties, most of which are discussed in the text. However, the approach seems reasonable though not well described (see below). Fortunately, the corrections do not have an excessively large effect on the reported yields (see Table 2), and the general conclusion that the 2HPr + acetyl peroxy route is about twice as important as the methyl glyoxal + acetaldehyde route is not affected. This is an interesting result in the context of SAR development, as well as for improving models for 3P2 reactions.

Response: The sentence "Fortunately, the corrections do not have an excessively large effect on the reported yields" suggest the possibility of a misunderstanding. The secondary processes for which the corrections were made have a certain antagonistic effect on the net formation of a product, which is reflected in the modelled yield values.

Comment 1c: Although this paper makes a contribution to the literature and should eventually be published, it does have some problems that need to be addressed. Most of the problems with this paper were noted in the posted review of Anonymous Reviewer #1, which I have read prior to writing my comments (but after reading the paper), along with the response by the authors. I agree with the comments of this reviewer, and believe that for the most part (but see below) the authors propose changes that should adequately address these comments.

Response: We understand that the reviewer read the comments of Anonymous Reviewer #1 and our answers before making his own comments. We wonder if the purpose Reviewer #2 in mentioning this, was to avoid duplicating the issues raised by Reviewer #1. Sometimes he describes the comments of Reviewer #1 and the corresponding answers without making a clear statement. An example is: "I […] believe that for the most part (but see below) the authors propose changes that should adequately address these comments". Does the referee agree or not with the changes we made? If he disagrees, we would kindly ask him for a clear statement. Further, in our understanding "[…] changes that should adequately address these comments" does actually mean that they DO NOT adequately address these comments.

Comment 2: The major problem noted by Reviewer #1 is the inadequate description of the modeling method used to correct the data and obtain the yield parameters. As the reviewer noted, the reference to the "Euler-Cauchy" model to derive parameters or corrections to the data is unclear, since this refers to a general solver method that could be applied to any system. In response, the authors improved the text around line 120 to better describe how the corrected yields were derived. The new Table 2 is a valuable addition to the text.

Response: The comment denotes a certain misunderstanding caused by the use of "model" and "method" in the original text. Reviewer #1 asked for more details on our approach, but he also asked what a "Euler-Cauchy-Approach" is (which has probably been a misunderstanding). On the other hand, what does "since this refers to a general solver method that could be applied to any system" mean? This is actually what has been done! Besides, we cite a previous work from our group where a description of the method is given.

Comment 3: However, the proposed changes to the reference of "Euler-Cauchy" in the abstract is not totally adequate. In responding to the reviewer, they changed the sentence in the abstract from "Employing an Euler-Cauchy model to describe the temporal behaviour of the experiments, the further oxidation of 2HPr was shown …" to "Employing a simple modeling tool describe the temporal behaviour of the experiments, the further oxidation of 2HPr was shown …". However, the modeling tool did not describe the temporal behavior of the chemical system, their assumed mechanism (given on the new Table 2) did. A better change may be "Employing a simple chemical mechanism to analyze the temporal behaviour of the experiments, the further oxidation of 2HPr was shown …". The modeling methodology need not be given in the abstract, since it is not newly developed in this study, and is presumably adequate.

Response: With all due respect, but the abstract should contain information on how the results were derived. The methodology is not given in the abstract, it is just said that branching ratios of the product were derived from modelling. Besides, since employing a simple chemical mechanism is exactly what we do with our simple modelling approach, the referee's suggestion would not change the content of this sentence. What does "presumably adequate" mean? The methodology can be either adequate or not. If not, the referee should provide a comment on that.

Comment 3: Reviewer #1 also noted that reaction of the hydroxy-substitued Criegee intermediate shown on Figure 2, forming acetaldehyde. is not the expected pathway. In any case, Figure 2 doesn't show the whole mechanism for the CI decomposition, since not all atoms are accounted for. In response, the authors wrote that acetaldehyde formation was unexpected and may be due to experimental impurities, and that speculation of the CI mechanism was beyond the scope of this work. But at least they should give a complete proposed mechanism on Figure 2, or show a "?" and "+ other products" on the figure to indicate that they don't know the mechanism.

Response: Once again, the referee is summarising a comment of Reviewer #1 and our response. With due respect, this comment shows that RC2 possibly did not read carefully the RC1's comments nor the paper. We DID NOT say that acetaldehyde formation may be due to experimental impurities. This has been a suggestion from Reviewer #1 and we explained why this is very unlikely. Fig 2 does not show the whole mechanism. This has never been claimed and it is not necessary to do so. Fig 2 does show the details necessary to understand how we derived a cross section for 2HPr. Given that acetaldehyde can only be formed from the lager CI, one cannot use 3P2OL for the cross section determination since there are two pathways forming acetaldehyde. We gave some thoughts to an in-depth analysis of the $O_3$ reaction, including all $RO_2$ reactions possible in the

experimental system, and this would deliver enough material for a paper of its own. Therefore, it is clearly beyond the scope of this study to include a whole mechanism.

Comment 4: However, there is quite a reasonable explanation for acetaldehyde formation from this CI on Figure 2. The H on the -OH could move to the outer O on the intermediate via a 6-member ring transition state, which could then rearrange rapidly to form acetaldehyde + formaldehyde + OH. This may not be the only route (the other route forming methyl glyoxal as the reviewer noted could also occur), but it is possible acetaldehyde formation may occur at least part of the time (and maybe dominate). Perhaps add both routes to Figure 2 and state in the discussion that relative importance of each is uncertain and not further investigated in this work.

Response: If the H atom of the OH-group were abstracted, this could not result in formaldehyde since there is only one H atom at the corresponding carbon atom. We agree that based on the current knowledge of CI chemistry one would also expect methyl glyoxal as a product. However, we could not confirm its formation in our experiments. On the other hand, the referee does also suggest that acetaldehyde is formed from the larger CI. This is the relevant information. Therefore, we do not see the necessity to include further speculations since this would not change our conclusions.

Comment 5: Reviewer #1 noted other problems with the paper, but it appears that the authors' responses were adequate, so I will not discuss them here. However, I have a few additional comments and suggestions. Other than these, I did not see major additional problems with the paper.

Response: We do not know how to understand this comment. What does "it appears that the authors' responses were adequate" mean? Either they are adequate or the referee should comment on that, if not. Moreover, Reviewer #1 has to judge if our responses to his comments are adequate.

Comment 6: It was unclear whether the product concentrations plotted on Figure 4 have been corrected for secondary reactions when that figure was first introduced. Later (around line 314) it is implied that Figure 4 shows uncorrected data. It might be a good idea to show both corrected and uncorrected data on Figure 4, so one can get a feel of the size and effects of the corrections.

Response: When Fig. 4 is first introduced (lines 229-230) it is said that 3P2 is corrected for the wall loss whereas no corrections were mentioned for the quantified products. Besides, the axis' inscriptions in Fig. 4 include the index "corr." only for 3P2. This all implies that no corrections were performed on the products' data. However, we understand that at least when introducing the figure this should be clearly pointed out. We therefore added in the parentheses in line 230:

"Plots of the identified products (in ppmV without corrections) […]"

On the other hand, inclusion of corrected product data in Fig. 4 would presume that we calculated the corrected mixing ratio for each data point. This is not the way we performed corrections. As written in the manuscript, the first-generation yields were obtained through modelling of the temporal behaviour of each quantified species, given that beside 2HPr all other reaction products have secondary sources in the experimental system. Therefore, we prefer to keep Fig. 4 as it is.

Comment 7: Around line 294 it was stated that yield plots of 2HPr showed "small but precise" curvature. However, the 2HPr data on Figure 4 don't look particularly curved. Are they referring to corrected data? If so, this is another reason to include corrected data on Figure 4.

Response: The referee is missing an important point since we have written "a small but precise curvature IN EACH EXPERIMENT". When plotting each experiment separately, the plot for 2HPr is precisely non-linear. But all experimental data are shown in one plot as written in the figure caption. Besides, the data from 480 L chamber were scaled to fit within the scale of the 1080 L data. Due to the statistical variation the non-linearity might be nearly invisible when combining all data. We had to find a middle way between giving all information without overloading the figure.

To avoid misunderstandings we modified the paragraph (lines 294 – 295) as follows:

"On the other hand, the yield plots of 2HPr show a small but precise curvature when the data set  of each single experiment is plotted separately. This effect becomes nearly invisible in Fig. 4 due to statistical scattering when combining all data and scaling the data of the 480 L chamber experiments into the scale of the 1080 L chamber experiments."

Comment 8: The failure to observe the PAN analogue CH3-CH(OH)-CO-OONO2 shown in Figure 7 might be due to another rapid 1,4 H-shift reaction where the H on the -OH moves to the peroxy O next to -NO2, forming HNO3 and (ultimately) CO2 and acetaldehyde. This could be potential source of acetaldehyde in the system that is not accounted for in their model (new Table 2). Would this affect the acetaldehyde yield from 3P2 that fit the data?

Response: This is a very interesting point describing a potential additional loss process of the PAN analogue, besides the thermal decomposition. However, the referee is describing an 1,5 H-shift (instead of 1,4 H-shift), which is expected to be much slower than a 1,4 H-shift. Since no information on the thermal stability of the PAN analogue is available, any further statement on a competition between an H-shift induced decomposition and the classical thermal decomposition would be highly speculative. On the other hand, while the H-shift reaction would yield, as the referee suggested, acetaldehyde and $CO_2$, the thermal decomposition would recycle the $RO_2$ radical which, in turn, yields acetaldehyde and $CO_2$ as well. Therefore, the modelled time profiles and thus the obtained branching ratios are independent from whether the PAN analogue species is just thermally unstable or additionally decomposes through the H-shift reaction.

---

## Author Comment (AC3)

Authors' responses to comments from Aparajeo Chattopadhyay on: Illmann et al., Atmos. Chem. Phys. Discuss., https://doi.org/10.5194/acp-2021-575

We thank the referee for the comments on this work. The original comments are shown in black and our responses are marked in blue. Changes made in the text are marked in red.

**General comments:**

This study by Illmann et al. describes kinetics, products study, and reaction mechanism for the OH-initiated oxidation reaction of 3-penten-2-one which had been detected in biomass burning plumes. To understand biomass burning chemistry and its role on tropospheric ozone and SOA formation, the chemistry of individual chemical components of a biomass burning plume should be known. This study presents a comprehensive understanding of the atmospheric chemistry of 3-penten-2-one which could be important from biomass burning perspective. The authors also discussed OH reaction for 2-hydroxypropanal which is an oxidation product of 3-penten-2-one.

This work is of high scientific quality and is suitable for readers of Atmospheric Chemistry and Physics (ACP). However, in many cases, the detail about the measurements and the analysis is missing in the manuscript which could bring questions to the mind of the readers. Therefore, I recommend publishing this manuscript in ACP after addressing the issues mentioned below. Necessary modification to the manuscript should be done accordingly.

Response: We thank the referee for the examination of our work.

Specific comments:

Comment 1: 1. Page 4, line 117: A polynomial function was used for CO2 calibration. Any possible reason why a linear correlation was not observed?

Response: This is a common behaviour observed for small molecules like NO, CO and CO2 when working with FTIR spectroscopy. These instruments are usually operated at a resolution of 1 cm-1. The absorptions of the above mentioned molecules consist of a resolved line spectrum (= discontinuous spectrum) where the energy difference of different transitions between rotational levels are << 1 cm-1. Therefore, a single data point reflects an average of several absorption lines. It is generally possible to obtain a calibration function for the integrated absorption. However, due to the fine structure of the absorption features one has to calculate different calibration functions for different sizes of the integrated absorption. Accordingly, a linear behaviour is only observed for very low integrated absorptions.

Comment 2: 2. Page 4, line 121: Describe Euler-Cauchy approach. At least some details are required for the readers of this manuscript who are not familiar with this method.

Response: Considering also the comments of Anonymous Reviewer #1 we understand that the way we used the terms "Euler-Cauchy approach" or "Euler-Cauchy model" was confusing. On the one hand, we noticed that in contrast to German the common English term is "Euler method" and we changed this accordingly in the

whole manuscript. On the other hand, the method describes only the most basic numerical procedure to solve differential equations. Our simple model, used to describe the temporal behaviour of the quantified species, can be basically written down in calculation programs like Microsoft Excel. Therein, the branching ratios (corresponding to the first-generation yield) of the simplified reaction scheme are included as parameters and varied until the simulated profile matches the experimental data. To avoid misunderstandings we modified the brief model description (page 4, lines 120–124) as follows:

"In order to obtain first-generation yields These yields were corrected for (= yields without impact of secondary reactions in the experimental set-up) using the Euler-Cauchy the temporal behaviour of all quantified species was simulated following the approach previously outlined in the recent literature (Illmann et al., 2021b). Here, the differential equations are constructed based on the simplified reaction sequence of each species and solved by the Euler method using calculation programs like Microsoft Excel. Input parameters are the rate coefficients of each of the sequence's reaction, the initial concentration of 3P2 and a time-dependent OH-concentration calculated based on the 3P2 consumption. The molar formation yields for products of the target reaction are included as variable parameters to be modified varied until the simulated temporal behaviour of each species matches the experimental data. The details are provided within Sect. 3.4."

**To avoid confusion, the abstract was modified as follows:**

"Employing an Euler-Cauchy model a simple modelling tool to describe the temporal behaviour of the experiments, the further oxidation of 2HPr was shown ..."

Comment 3a: 3. Page 5, line 147: The sample purity is low. Even a sample purity of 85% is not good in this kind of precise measurement and it could severely impact the product studies. It creates a lot of uncertainty if we don't know what is that 15% impurity and what chemistry it can bring here.

Response: See below (Response to 3a, 3c, 3d)

**Comment 3b: Did you try to distill the sample?**

Response: No. But if purification could be easily achieved through distillation, the suppliers would offer samples with purities higher than technical grade.

Comment 3c: Given the fact that the IR cross-section of 3-penten-2-one was determined in the present study, it is not possible to check impurities using FTIR if you don't have reference spectra of the sample from other independent studies that used more pure samples.

**Response: See below (Response to 3a, 3c, 3d)**

Comment 3d: FTIR itself is not sensitive enough to identify small impurities and a more sensitive method e.g., GCMS could have been used.

Response to 3a, 3c, 3d: As written in the experimental section, we compared spectra of the 85% sample with spectra of a 70% sample where the impurity consists almost exclusively of mesityl oxide (= 4-methyl-3-penten-2-one). FTIR spectra of the 70% sample after subtraction of mesityl oxide were found to be identical to the

FTIR spectra of the 85% sample. Therefore, it is quite certain that the absorption features present in the spectra of the 85% sample belong to 3P2 exclusively. The integrated absorption signals used for the cross section determination are therefore unlikely affected by impurities. Maybe our statement in lines 148 – 149 was not precise enough since we have written that the 70% sample contains "mainly 4-methyl-3-penten-2-one as impurity". However, the supplier states that the sample contains 30% 4-methyl-3-penten-2-one. We therefore modified the sentence as follows:

"Another sample of 70% purity (technical grade) contains, as stated by the supplier, mainly 30% 4-methyl-3-penten-2-one as impurity."

We agree that impurities, if organic, could perhaps be identified with GC-MS. However, this technique was not available.

The match between spectra of the 85% sample and the 70% sample, after subtraction of 4-methyl-3-penten-2-one, indicates that there is no single compound dominating within the impurities of the 85% sample. Any effect on the observed and quantified reaction products would presume that 1) the impurities would either photolyze or react with OH with rates that are similar to the experimentally observed loss rates of 3P2, and 2) a large fraction of the impurity and hence several compounds would yield the same reaction products as 3P2. This is very unlikely.

Comment 4a: 4. Page 6, line 154: Why is the loss rate of the sample about one order of magnitude larger in the chamber having higher volume? I expected an opposite trend as the S/V is expected to be less for the larger cell.

Response: This behaviour is quite normal in our chambers. In this case, the wall losses are rather affected by the wall properties than by the S/V ratio. In contrast to borosilicate glass (480 L chamber) the surface of quartz glass (1080 L chamber) is acidic which sometimes causes much larger wall losses.

Comment 4b: Please show representative first-order decay plots for wall loss in Supplement.

Response: We do not see any benefit from including representative first-order decay plots for the wall loss since these were determined in each single experiment, as written in the manuscript, and vary from day to day depending on the wall properties. Time-profiles for product studies conducted in both chambers were given in Figs. 8 and 9 (of the originally submitted manuscript). These do clearly show the different behaviour of 3P2 in the dark and in our opinion this is sufficient.

**Comment 4c: What are the S/V values for the two chambers?**

Response: As mentioned above, the S/V ratios are not responsible for the different behaviour towards the chamber wall.

Comment 4d: What are the wall loss rates for the reference compounds?

Response: The reference compounds did not show any wall loss. For clarity, we included this in lines 154 – 155:

"The first-order wall loss of 3P2 was <  $1 \times 10^{-5}$  s-1 in all 480 L chamber experiments and in the range of (5–10)  $\times 10^{-5}$  s-1 in the 1080 L chamber experiments, respectively. Typically, the consumption through OH radicals was about one order of magnitude faster than the wall loss. The reference compounds did not show any wall loss.

Comment 5a: 5. Page 6: 3-penten-2-one + OH kinetic data looks satisfactory. For presentation purposes, the data points for individual experimental runs could be shown using different symbols/colors to highlight data quality for individual experimental runs.

Response: We thank the referee for this suggestion and updated Fig. 1 and the corresponding figure caption as follows:

**Figure 1.** Relative-rate plots of all experiments using isoprene (green) and E2-butene (blue) as references. Different experimental runs for each reference are denoted with different symbols. The error bars consist of a systematic uncertainty and an additional 10% relative error to cover uncertainties derived from the experimental and evaluation procedure, respectively.

Comment 5b: Typically, how much corrections (in percentage) were made on individual data points to account for the 3-penten-2-one wall loss?

Response: Typically, < 10%.

Comment 5c: It was noted that a previous study from the same laboratory reported a rate coefficient that is slightly higher than the number obtained in the present study. It was noted that they agree within 20%, to my opinion this difference is rather large. Mesityl oxide was an impurity in the previous study, but this should not bias kinetic results in relative rate measurements if it does not interfere with your spectral subtraction. In fact, when I see kinetic data for Isoprene reference (Figure 1), there are some points (probably correspond to the last row of Table 1 i.e., Expt 3P2#6) that are bias low making the whole kinetic data scatter and the rate coefficient bias low. If only E2-butene data are considered, then the correspondence is much better ~ 10%.

Response: The reviewer observed correctly that there was one experiment, when isoprene was used as the reference compound, yielding slightly lower values. However, we could not find any reason to exclude this experiment from the rate coefficient calculation. The observation that "if only E2-butene data are considered, then the correspondence is much better ~ 10%" is true but might just be coincidence. For the rate coefficient determination it is necessary to perform a statistically relevant number of experiments and hence one should

always avoid working with only one reference compound. However, analyzing the data shows they are no outliers. We do not completely understand the statement of referee #3 since, considering only half of our results would end up in optimizing the results, which, for sure, was not intended by the reviewer.

It is the reviewer's personal opinion, as stated by himself, that a difference of 20% is "rather large" in between the two rate coefficient determinations. In our opinion, a difference of 20% is quite good, given that even IUPAC recommendations do not seldomly exceed uncertainties of 20% for species where many datasets are available. The additional mesityl oxide and consequently its reaction products add complexity to the reaction system and the FTIR spectra, which could theoretically interfere with the subtraction procedure. However, it is not our purpose to judge the previous measurements. Moreover, the statement was based on comparing the average values solely. We should have mentioned that considering the uncertainties both values are the same. To avoid confusion, we modified the sentence (line 172) as follows:

"Nevertheless, both determinations agree within 20% and the value obtained here is within the uncertainties of the former study."

Comment 6a: 6. Page 7: In situ generation of 2-hydroxypropanal – Determination of 2-hydroxypropanal (2HPr) infrared cross-section, which was used later for its yield determination, is based on the assumption Yield (2HPr) = 1 -Yield (HCHO). This assumption is valid if HCHO is formed only by pathway a (Figure 2) and 2HPr is formed only by pathway b and there are no other pathways for POZ decomposition. The authors did not discuss the total fate of corresponding CIs (from pathways a and b) which can complicate the analysis. For example, the CI for pathway b for 3B2OL is CH2OO i.e., the simplest Criegee Intermediate. Bimolecular self-reaction of CH2OO (whose rate is very fast) could form additional HCHO, then the above assumption would be invalid. Similarly, it was shown that the branching ratio for acetaldehyde formation from the CI that is produced from pathway a is 0.36, what is the fate of the rest of CI? If there is a bimolecular self-reaction, then it would produce additional 2HPr. Give proper evidence that these bimolecular self-reactions of CIs are not happening in your experimental condition.

Response: Based on our current understanding of  $O_3$  reactions there are no POZ decomposition pathways other than the two decomposition routes yielding one or the other primary carbonyl and the remaining Criegee Intermediates. A fraction of the  $O_3$  + 3B2OL reaction could hypothetically yield an epoxide. As already mentioned in the manuscript, there is no hint for epoxide formation. We did not discuss the whole fate of the Criegee Intermediates since deciphering every reaction channel and possible  $RO_2$  reactions in the experimental system is clearly beyond the scope of this study.

The reviewer suggests secondary sources of both primary carbonyls through self-reaction of the Cls. Bimolecular reactions of Criegee Intermediates are, on the one hand, limited to the fraction of stabilised Cls since the other Cl fraction undergoes prompt decomposition or isomerisation. On the other hand, even if the bimolecular rate coefficient of the  $CH_2OO$  self-reaction is in the order of  $10^{-11}$  cm3 molecule-1 s-1, the overall rate (the bimolecular rate coefficient times the concentrations) determines whether this reaction can compete with other bimolecular reactions or not. For example, traces of formic acid (ppb levels) were present in the experimental system. Given that sCl levels are well below ppb levels in our experimental system and the rate coefficient of  $CH_2OO + HC(O)OH$  is even a factor of 1.5 larger (IUPAC, current recommendation), bimolecular self-reactions of sCls can be ruled out.

Comment 6b: The quoted uncertainty for OH absorption band cross-section is very high which would make the product yield data unreliable. The high uncertainly was chosen for the error on the HCHO yield and wall loss of 2HPr. Give more details on that.

Response: We do agree that the marge of error is quite large. However, we do argue that, given the indirect method used to determine the cross-section of 2HPr, this value is quite good. As written in the manuscript the cross section determinations agree within 9% within different experiments, which hence includes the uncertainty derived from the wall loss of 2HPr. The uncertainty for the HCHO yield was already given as well in the manuscript. Combining the uncertainties yields a relative error below 30%. However, we extended the uncertainty considering the fact that the cross section determination is performed in a complex chemical system.

**Comment 6c: Does the wall loss follow the first-order decay?**

Response: Yes.

Comment 6d: Did you also observe wall loss for 3B2OL and corrected it?

Response: The wall loss for 3B2OL was  $< 2 \times 10^{-5}$  s-1 and hence negligible.

Comment 7: 7. Page 11, line 266: If the loss of acetaldehyde and methylglyoxal by OH reaction is significant, should not you observe nonlinearity in their yield data (Figure 4) at longer times?

Response: This is an interesting point! The plots for both aldehydes are linear. But this is coincidence since secondary formation and loss processes almost compensate each other within the experiments investigating the 3P2 + OH reaction. One should note that it is not possible to simulate the time profiles for both aldehydes without considering the secondary formation due to 2HPr + OH. The non-linearity would become visible with much longer irradiation times.

Comment 8: 8. Figure 5: The fate of CH3CO radical following reaction with O2 and NO is shown to be the formation of CO2 and HCHO. If HCHO is formed by CH3OO reaction, then other products such as methanol are also expected.

Response: This is not true for the conducted experiments. Methanol would be formed from the self-reaction of the methylperoxy radical. As written in the manuscript, all experiments were conducted under conditions where all  $RO_2$  radicals will react with NO. Thus, the CH3OO radical is converted solely to the CH3O radical, which results in formaldehyde.

Comment 9: 9. Page 13, line 312: Describe the model that was used to correct 2HPr yield due to the 2HPr + OH reaction. Input/output could be provided as a Supplement.

Response: The brief description of the model was updated as written in the response to comment 2. For clarity we added as well an additional table listing the simplified reaction sequence needed to describe the reaction system. This includes the reaction scheme as well as the input (kinetic) data and the parameters (first-generation yields) derived from the modelling. Given that our approach does not fully describe the mechanism, the obtained branching ratios for the reactions listed in the table 2 represent the first-generation yields and not exact branching ratios in the case of 3P2 + OH. For example, the average first-generation yield of acetaldehyde obtained through modelling is 0.38, which corresponds to the branching ratio R4b/R4 according to the simplified reaction scheme. However, given that according to our proposed mechanism acetaldehyde can be formed from addition to both  $C_{\alpha}$  and  $C_{\beta}$ , one can just conclude that 38% acetaldehyde are formed from 3P2 + OH. Since it is not possible to decipher the branching ratios. By contrast, in the case of 2HPr, the obtained first-generation yields correspond to the branching ratios for the different abstraction channels.

| Reaction |                                                          | Branching | Rate coefficient                                                     |      |
|----------|----------------------------------------------------------|-----------|----------------------------------------------------------------------|------|
|          |                                                          | ratio     |                                                                      |      |
| R4       | $CH_3CH=CHC(O)CH_3 + OH \rightarrow products$            |           | $6.2 \times 10^{-11} \text{ cm}^3 \text{ molecule}^{-1} \text{ s}^1$ | a, b |
| R4a      | $\rightarrow$ CH 3 C(OH)C(O)H                 | 0.68 °    |                                                                      |      |
| R4b      | $\rightarrow$ CH 3 C(O)H                      | 0.39 °    |                                                                      |      |
| R4c      | $\rightarrow$ CH 3 C(O)C(O)H                  | 0.32 °    |                                                                      |      |
| R4d1     | $\rightarrow$ CH 3 C(O)OONO 2      | d         |                                                                      |      |
| R4d2     | $\rightarrow$ CO 2 + HCHO                     | d         |                                                                      |      |
| R5       | $CH_{3}CH=CHC(O)CH_{3} + wall \rightarrow$               |           | $\leq 1.0 	imes 10^{-4} 	ext{ s}^{-1}$                               | a, e |
| R6       | $CH_3C(O)H + OH \rightarrow products$                    |           | $1.5\times10^{\text{-}11}\text{cm}^3$ molecule^-1 $s^1$              | a, f |
| R6a      | $\rightarrow$ CH 3 C(O)OONO 2      | a, g      |                                                                      |      |
| R6b      | $\rightarrow$ CO 2 + HCHO                     | a, g      |                                                                      |      |
| R7       | $CH_3C(O)H + wall \rightarrow$                           |           | $\leq 1.0 	imes 10^{-4} 	ext{ s}^{-1}$                               | a, e |
| R8       | $CH_3C(O)C(O)H + OH \rightarrow products$                |           | $1.3\times 10^{\text{-}11}\text{cm}^3$ molecule^-1 $s^1$             | a, h |
| R8a      | $\rightarrow$ CO + CH 3 C(O)OONO 2 | i         |                                                                      |      |
| R8b      | $\rightarrow$ CO + CO 2 + HCHO                | i         |                                                                      |      |
| R9       | $CH_3C(O)C(O)H + wall \rightarrow$                       |           | $\leq 1.0 	imes 10^{-4} 	ext{ s}^{-1}$                               | a, e |
| R10      | $CH_3C(OH)C(O)H + OH \rightarrow products$               |           | $1.7\times 10^{\text{-}11}\text{cm}^3$ molecule^-1 $s^1$             | a, j |
| R10a     | $\rightarrow CO_2 + CH_3C(O)H$                           | 0.79 °    |                                                                      |      |
| R10b     | $\rightarrow$ CH 3 C(O)C(O)H                  | 0.21 °    |                                                                      |      |
| R11      | $CH_3C(OH)C(O)H + wall \rightarrow$                      |           | $\leq 1.0 \times 10^{-4} \text{ s}^{-1}$                             | a, e |

**Table 2.** Simplified reaction sequence used for the modelling of the temporal behaviour of experimentally quantified species in order to obtain first-generation yields for the respective reactions (3P2 + OH, 2HPr + OH).

a Input parameter; b rate coefficient determined within this work; c average branching ratio obtained through modelling; d average branching ratio (R4d1+R4d2)/R4 = 0.58 obtained through modelling; e determined in each individual experiment; f rate coefficient from Atkinson et al. (2006); g branching ratio (R6a+R6)/R6 = 0.95 from Atkinson et al. (2006); h rate coefficient from Atkinson et al. (2006); i branching ratio (R8a+R8b)/R8 = 1.0 from Atkinson et al. (2006); j rate coefficient from Baker et al. (2004).

Comment 10: 10. Page 13, line 330: A rate coefficient for 2HPr + OH reaction is quoted but the details on how this number was obtained are not presented. Please provide the necessary details. The rate coefficient differs from a previously measured value by 30% which is not an excellent agreement.

Response: As already written in the manuscript, the rate coefficient was estimated based on the method used previously by Baker et al. (2004). In this study the rate coefficients of in situ generated hydroxycarbonyls were obtained by a non-linear regression analysis of a plot of the formed hydroxycarbonyl vs. ln([precursor]0/[precursor]t). For this procedure, it is not necessary to include the first-generation yield of the in situ generated species since the plots profile depends only on the rate coefficient ratio k(hydroxycarbonyl + OH)/k(precursor + OH). This method is applicable only if the target species has no or at least no significant wall loss. Hence, only a very limited number of experiments allowed us to apply this approach and we therefore prefer to consider the value obtained here as an "estimated rate coefficient", as written in the abstract, and to assign an expanded uncertainty. We understand, however, that how we derived k(2HPr+OH) was not clearly written and we would therefore include an additional figure showing the non-linear plot following the procedure outlined by Baker et al. (2004):

Figure 8. Non-linear plot for the formation of 2HPr from the 3P2 + OH reaction according to Baker et al. (2004) used to estimate the rate coefficient of 2HPr + OH.

**We further modified the corresponding paragraph (page 13, lines 329–332) as follows:**

"Applying the approach presented by Baker et al. (2004) to these experiments allows to estimate <del>yields</del> a value of (2.2 ± 0.6) ×  $10^{-11}$  cm3 molecule-1 s-1 for the rate coefficient of 2HPr + OH <del>which is about 30% larger than</del> previously reported. The non-linear plot, drawn according to the previously mentioned approach, is presented in Fig. 8. Our estimation is about 30% larger than previously reported (Baker et al., 2004). <del>Given</del>Taking into account that both determinations are based on the in situ generation of the  $\alpha$ -hydroxyaldehyde this is still an excellent agreement."

Given that both determinations are based on the in situ generation of the aldehyde and thus the rate coefficient is determined from a complex chemical system, this is, in our opinion, an excellent agreement. Moreover, within the uncertainties assigned to our value, which is stated as an estimated value, both determinations are the same.

Comment 11a: 11. Page 14, line 353: Again, the details of the model that was used to find the branching ratios, was not given. Provide necessary details here and Input/Output as a supplement.

**Response: See above.**

**Comment 11b: Did you do any sensitivity analysis?**

Response: Yes, we did. Our conclusions, the given branching ratios as well as their assigned uncertainties are based on a careful sensitivity analysis. Based on the suggestion from Anonymous Reviewer #1 we updated Fig. 9 (Fig. 8 in the originally submitted manuscript) in order to show sensitivity analysis and included the temporal behaviour of acetaldehyde and methyl glyoxal, assuming different branching ratios for the 2HPr + OH reaction. We also included explicitly the sensitivity analysis in the corresponding paragraph (page 14, lines 346–360):

"Molar formation yields for acetaldehyde and methyl glyoxal, derived from 3P2 + OH, and the branching ratio  $k_b/[k_a + k_b]$ , resulting in methyl glyoxal through 2HPr + OH, were included as parameters in the simplified model (Illmann et al., 2021b) and varied until the experimental time profiles are reproduced by the simulation. Since the OH initiated oxidation of 2HPr is expected to proceed solely through (a) and (b) the branching ratio  $k_a/[k_e + k_b]$  is given as  $1 - k_b/[k_e + k_b]$  (Fig. 7). Table 2 shows the simplified reaction sequences and the rate coefficients needed to describe the reaction system. These sequences do not follow the nomenclature used in the proposed mechanisms (Figs. 5 and 7) since the simplified model does not differentiate if a product is formed directly from a parent compound through more than one pathway.

Figure 9 summarises an analysis of the model sensitivity on the branching ratios  $k_a$  and  $k_b$ . As can be seen in panel (a), a variation from  $k_a = 1.00$  and  $k_b = 0.00$  (sim1) to  $k_a = 0.50$  and  $k_b = 0.50$  (sim3) exhibits no measurable influence on the temporal profile in the beginning of the first irradiation period, corresponding to a 3P2 consumption of

---

## Author Response (AR2)

Authors' responses to comments on: Illmann et al., Atmos. Chem. Phys. Discuss., https://doi.org/10.5194/acp-2021-575

We thank the referees for the additional comments on this work. The original comments are shown in black and our responses are marked in blue. Changes made in the text are marked in red.

**A. Comments by Referee 1**

Comment 1: In one of my comments, I stated that the abstract should say they used a simple chemical mechanism to describe the experiments, rather than a "modeling tool". Rather than address it, the author chose to criticize my comments and apparently misunderstood this rather obvious point. A "modeling tool" is just a software and/or algorithm, and doesn't describe any chemistry. It is the chemical mechanism implemented in the tool that describes what is happening, and it could be employed using any number of modeling tools. The author seems to think that I don't know the difference between a modeling tool and a mechanism, but after over 40 years working in this field I suspect the problem is more likely with him.

Response: First of all, we would like to emphasize that we never intended to question the expertise of the referee. We understand the referee's point that a software does generally not describe any chemistry and that our wording is not entirely correct. However, the simple chemical mechanism itself does not contain any number. It just says that acetaldehyde and methyl glyoxal are formed through the 2HPr oxidation. The branching ratios are obtained through modelling. Therefore, in our opinion, the referee's suggestion did not completely solve the raised issue either. Nevertheless, we understand that the referee's suggestion is a better wording in order to avoid misunderstandings. The sentence was modified accordingly.

Comment 2: A comment I made regarding the apparent formation of acetaldehyde in the O3 + 2BOL reaction, was also not satisfactorily addressed. Figure 2 shows acetaldehyde coming from the OH-substituted Criegee, but doesn't show a co-product or suggest a mechanism, since its formation is not expected based on what is known of simpler Criegees. I gave a suggested mechanism but I incorrectly wrote "formaldehyde" when I should have written "formyl radicals" (ultimately giving acetaldehyde + CO + HO2 + OH). The authors correctly noted my error, but apparently didn't bother to look at the system and see that the co-product should obviously be formyl. Their main response was that the main point is that acetaldehyde is probably formed from this Criegee (or so the data suggest), but speculation on the mechanism was beyond the scope of the paper. This is a fair point, but in this case, they should not pretend it is a mechanism on Figure 2, but instead should show a "?" and "+ other products" as I suggested in my comment.

Response: We still do not understand this issue since we do not pretend it is a complete mechanism but only a more likely reaction sequence. It is just said that acetaldehyde formation originates from the larger CI. However, in order to make unambiguously sure that the CI further chemistry yields additional unidentified products, we added reaction pathways yielding "products", although we believe that this was obvious from our discussion. Additionally, we replaced the biradical structures of the CIs with zwitterion structures to be consistent with the recommendation by the latest IUPAC Review article (Cox et al., ACP 2020, https://doi.org/10.5194/acp-20-13497-2020).

**Figure 1.** Formation of 2-hydroxypropanal through the ozonolysis of 3-buten-2-ol (3B2OL, red) and 3-penten-2-ol (3P2OL, blue) and respective average branching ratios. For readability reasons only one stereo-isomer is drawn for each Criegee Intermediate.

Comment 3: I also commented on their statement that there was curvature in Figure 4 in the yield plots for 2HPr, but this was not evident in the figure. In response, the author stated that there was curvature for individual experiments, but this is not evident in Figure 4 because it shows all the experiments with a single symbol. Since this curvature is noted in the text, the readers should be able to see it for themselves. Either use different symbols for different experiments in Figure 4 or, if this doesn't show this clearly, then show a plot of a representative experiment as an insert or in a supplement.

Response: We modified Fig 4 as follows: 1) different experimental runs are denoted with different symbols, 2) we deleted the regression line in panel (c), and 3) the colours were changed in order to use the same colour code throughout the manuscript. Additionally, we noted that at least for acetaldehyde, methyl glyoxal and PAN +  $CO_2$  the error bars included accuracy errors which is not useful when proving linearity or non-linearity of the plot. Accordingly, the figure was updated and the precision errors included. We hope that in this way it becomes visible for readers that the plots exhibit a high linearity for acetaldehyde and methyl glyoxal in contrast to 2HPr and the sum of PAN and  $CO_2$ . Note that, since based on a comment from Referee 2 some new figures were added, we added a Supplement. The former Fig. 4 becomes Fig. 3 in the revised manuscript.

**Figure 3.** Yield plots for (a) acetaldehyde, (b) methyl glyoxal, (c) 2-hydroxypropanal, and (d) the sum of PAN and  $CO_2$  for all conducted experiments corrected for the wall loss of 3P2. The error bars consist of the corresponding precision error. The data of the 480 L chamber experiments are multiplied with a factor of 0.1 to fit within the scale of 1080 L chamber experiments. Different experimental runs are denoted with different symbols.

Comment 4: With regard my proposed mechanism for forming formaldehyde from the PAN analogue shown on Figure 7, the author criticized my suggesting about what I called a "1,4-H" shift of the PAN analogue, stating it was really a 1,5-H shift, which is less favorable (but not always). But this suggested reaction actually has a 6-member ring transition state, so can't be criticized on this basis. However, there is no need for the authors to accept this suggestion because it does not affect the results of interest, though the possibility of another route for acetaldehyde formation would be of interest in the context of this figure.

Response: Our response did not intend to criticize the referee. But the referee described mechanistically an 1,5-H shift but wrote 1,4-H shift instead. This has just been pointed out in order to be sure that we did not misunderstand the comment. In his original comment the referee asked if this additional route affects the obtained yield for acetaldehyde. Given that both the thermal decomposition and the potential decomposition after the 1,5-H shift would finally yield acetaldehyde and  $CO_2$  we pointed out that this does not affect the results. We agree on the thought on the 6-member ring transition state. In this context, an 1,5-H shift is much faster than an 1,4-H shift in, probably, nearly all cases. However, we were thinking that in this particular case an 1,5-H shift should be much slower given that in RO2 radicals, for instance, rate coefficients for migration of hydroxyl-H atoms are very low (Vereecken, L. and Nozière, ACP 2020, https://doi.org/10.5194/acp-20-7429-2020). However, the PAN species is of course not a radical. But therefore, we would think an H shift reaction to be a sigmatropic rearrangement which requires a  $\pi$ -bonded system that rearranges during the shift reaction which is not the case. Given that we are not sure if a shift reaction is possible, we prefer to keep the figure as it is. In order to keep H shifts in mind, we added some words with respect to the 2HPr derived RO2 radical, given that theoretical calculations indicate fast H shifts in acylperoxy radicals:

"A theoretical investigation on the C5-acylperoxy radical indicates that H migration reactions (1,5-H, 1,6-H or 1,7-H shift) of larger acylperoxy radicals might be fast enough to compete with the bimolecular reactions at low ppb levels of NO (Knap and Jørgensen, 2017). However, based on the predicted effects of the substitution pattern on the reactivity towards H migration reactions (Vereecken and Nozière, 2020) one would not expect the unimolecular reaction of the smaller hydroxypropionylperoxy radical to be competitive, at least not under our experimental conditions."

**B. Comments by Referee 2**

Comment 1: The manuscript is significantly improved – in particular, the inclusion of Table 2 and Figure 8, as well as the additional information provided in the new Figure 9, is helpful. I have one point that should be addressed further, however: In Figure 9, uncertainties in the measured product mixing ratios (2HPr, CH3CHO, CH3COCHO) should be shown. I am guessing that the +/- 0.05 uncertainties on the ka and kb values are too 'optimistic'?

Response: Thank you for raising this point! Indeed, our analysis was focused on reproducing the entire profiles and, in this context, did only consider precision errors, although for the overall uncertainties accuracy errors of all species should have been considered. We therefore re-performed the sensitivity analysis with respect to the accuracy errors. We refined our model and considered the acetaldehyde and methyl glyoxal yield from 2HPr + OH separately (formerly the sum of both was set to 1). ka and kb are then obtained from normalization of the corresponding yields. In order to find the maximum uncertainty on the branching ratios  $k_a$  and  $k_b$  two scenarios were defined in which acetaldehyde is simulated for the upper limit of the accuracy error and methyl glyoxal for the lower limit, respectively, or the inverse case. These scenarios were simulated for a variety of 2HPr yields within the accuracy error. This procedure changed slightly the average branching ratios and the assigned errors. They are, however, identical to the former values considering the assigned uncertainty. In order to address this analysis properly, main parts of Sect. 3.4 were re-written and accuracy errors added to the corresponding figures. Given that the modelling part became much more predominant within this section, we changed the heading from "2-Hydroxypropanal + OH and yields correction" into "2-Hydroxypropanal and modelling". Additionally, the figures show simulations for the different scenarios. A similar analysis was performed for PAN +  $CO_2$ . Additional graphical representations of the sensitivity analysis were added to a new Supplement. The original Figs 3 and 8 were also shifted to the Supplement. The main text of Sect. 3.4 was modified as follows:

"Among the class of  $\alpha$ -hydroxyaldehydes mechanistic information on the OH reaction and photolysis at atmospheric pressure were reported merely for glycolaldehyde (Niki et al., 1987; Bacher et al., 2001; Magneron et al., 2005). Baker et al. (2004) reported rate coefficients for the OH reaction of a series of hydroxyaldehydes synthesized in situ via the reaction of OH radicals with precursor alcohols. The authors obtained  $k = (1.7 \pm 0.2) \times 10^{-11}$  cm3 molecule-1 s-1 for 2HPr + OH through a nonlinear least squares analysis of the data of the 2-methyl-2,4-pentanediol oxidation (Baker et al., 2004). Under tropospheric daytime conditions both photolysis and OH initiated oxidation are important removal processes for glycolaldehyde (Bacher et al., 2001). However, assuming a similar behaviour for 2HPr, photolysis is expected to be negligible under the experimental conditions of the present study, since the OH radical level is much higher while photolysis frequencies are lower than within the troposphere. Including the above rate constant of 2HPr + OH into the model described previously (Illmann et al., 2021b) results in a corrected average yield of 0.68 ± 0.27, which is about 15% higher than determined from the yield plot, without proper corrections (Fig. 3Fig. 4). Hence, a significant fraction of the  $\alpha$ -hydroxyaldehyde is subject to OH radical initiated oxidation.

Based on the SAR approach by Kwok and Atkinson (1995) and the mechanistic information reported for the glycolaldehyde oxidation (Niki et al., 1987; Bacher et al., 2001; Magneron et al., 2005) one would expect

abstraction of the aldehydic H atom to dominate compared to abstraction of the carbon-bonded H atom of the –CH(OH)– entity for the OH reaction of 2HPr, as presented in Fig. 6Fig. 7. The abstraction from the terminal  $-CH_3$  group and the -OH group is expected to be negligible due to the much lower group rate constants. The hydroxypropionyl radical formed according to channel (a) will either eliminate carbon monoxide and react with O2 to form acetaldehyde or react with oxygen to form a hydroxypropionylperoxy radical (Fig. 6). The latter radical, resulting from channel (2a), may either yield peroxyhydroxypropionyl nitrate or will be converted to the corresponding RO radical (Fig. 6Fig. 7). This species will readily eliminate  $CO_2$  and finally form acetaldehyde as well. A theoretical investigation on the C5-acylperoxy radical indicates that H migration reactions (1,5-H, 1,6-H or 1,7-H shift) of larger acylperoxy radicals might be fast enough to compete with the bimolecular reactions at low ppb levels of NO (Knap and Jørgensen, 2017). However, based on the predicted effects of the substitution pattern on the reactivity towards H migration reactions (Vereecken and Nozière, 2020) one would not expect the unimolecular reaction of the smaller hydroxypropionylperoxy radical to be competitive, at least not under our experimental conditions. By analogy to the OH initiated oxidation of 3-hydroxy-2-butanone (Aschmann et al., 2000) one would expect reaction with oxygen to predominate over decomposition for the hydroxyalkyl radical formed following channel (b), thus leading to methyl glyoxal (Fig. 6Fig. 7). Hence, the 2HPr + OH reaction appears to be is expected as a secondary source of acetaldehyde and methyl glyoxal in the experimental system.

In order to investigate the 2HPr + OH reaction, methyl nitrite and NO were added for a second time, after a 3P2 consumption of about 70%, to shift the reaction system towards secondary oxidation processes. Applying the approach presented by Baker et al. (2004) to these experiments, as shown in Fig. S2 in the Supplement, allows to estimate a value of  $(2.2 \pm 0.6) \times 10^{-11}$  cm3 molecule-1 s-1 for the rate coefficient of 2HPr + OH. The non-linear plot, drawn according to the previously mentioned approach, is presented in Fig. 8. Our estimation is about 30% larger than previously reported (Baker et al., 2004). Taking into account that both determinations are based on the in situ generation of the  $\alpha$ -hydroxyaldehyde this is still an excellent agreement. As shown in Figure 9, 2HPr (green) passes through a small maximum during the second irradiation period. In panel (c) of Fig. 7 it can be observed that the The-mixing ratio of acetaldehyde (purple circles) increases continuously over the second irradiation period, while that of methyl glyoxal (black circles) is reaching relatively fast a plateau. at the end of the reaction in the case of methyl glyoxal (black) This is in qualitative agreement with the proposed mechanism. Peroxy nitrates other than PAN, formed through 3P2 + OH, could not be detected. Traces of the analogue peroxyhydroxyacyl nitrate resulting from glycolaldehyde oxidation have only been previously observed when the corresponding RO2 radical was generated through the reaction of glycolaldehyde with Cl atoms in the presence of NO2 (Niki et al., 1987). Magneron et al. (2005) did not detect any PAN-type species in the glycolaldehyde + OH system and therefore concluded that this species is probably likely unstable and readily dissociates. Hence, abstraction of the aldehydic H atom following channel (a) will likely result exclusively in the formation of acetaldehyde irrespective of the branching ratio between the (1a) and the (2a) channel (Fig. 6Fig. 7).

Molar formation yields for acetaldehyde and methyl glyoxal, derived from 3P2 + OH, and the branching ratio kb/[ka + kb], resulting in methyl glyoxal through 2HPr + OH, were included as parameters in the simplified model (Illmann et al., 2021b) and varied until the experimental time profiles are reproduced by the simulation. Since the OH initiated oxidation of 2HPr is expected to proceed solely through (a) and (b) the branching ratio  $k_p$  is given as  $1 - k_p$  (Figure 7). Table 2 shows the simplified reaction sequences and the rate coefficients needed to describe the reaction system. These sequences do not follow the nomenclature used in the proposed mechanisms (Figs. 5 and 7) since the simplified model does not differentiate if a product is formed directly from a parent compound through more than one pathway.

Figure 9 summarises an analysis of the model sensitivity on the branching ratios ka and kb. As can be seen in panel (a) a variation from  $k_a = 1.00$  and  $k_b = 0.00$  (sim1) to  $k_a = 0.50$  and  $k_b = 0.50$  (sim3) exhibits no measurable influence on the temporal profile in the beginning of the first irradiation period, corresponding to a 3P2 consumption of < 30%. Hence, the secondary oxidation of both acetaldehyde and methyl glyoxal was negligible immediately after switching on the lamps for the first time. This allows to derive values for the firstgeneration yields of acetaldehyde and methyl glyoxal from 3P2 + OH in these experiments. The branching ratios (ka and kb) chosen for sim1 to sim3, depicted in panel (a) completely fail in reproducing the profile for both reaction products for the entire duration of the experiment. Panel (b) in Fig. 9 shows the optimum range for the branching ratios that allows a simultaneously fit of the experimental time profiles for both acetaldehyde and methyl glyoxal. Accordingly, the branching ratios ka and kb were found to be 0.79 ± 0.05 and 0.21 ± 0.05, respectively. This is in excellent agreement with SAR predictions (Kwok and Atkinson, 1995) which estimate 0.8 and 0.2 for the branching ratios, respectively, as well as former results on the OH reaction of glycolaldehyde at atmospheric pressure (Niki et al., 1987; Bacher et al., 2001; Magneron et al., 2005).

The molar formation yields of acetaldehyde and methyl glyoxal, derived from 3P2 + OH as well as from 2HPr + OH were included as parameters in a simplified model (Illmann et al., 2021b) and varied until the experimental time profiles are reproduced by the simulation. Since the OH initiated oxidation of 2HPr is expected to proceed solely through the channels (a) and (b), the product yields of acetaldehyde and methyl glyoxal, from 2HPr + OH, should correspond to the branching ratios  $k_a$  and  $k_b$ , respectively (Fig. 6). Their sum should, in turn, equal unity. Table 2 shows the simplified reaction sequences and the rate coefficients needed to describe the reaction system. These sequences do not follow the nomenclature used in the proposed mechanisms (Figs. 4 and 6) since the simplified model does not differentiate if a product is formed directly from a parent compound through more than one pathway.

Figure 7 summarises an analysis of the model sensitivity, observing also the accuracy of all quantified species. For all species but 2HPr accuracy was defined as a 10% relative error plus the corresponding detection limit. The accuracy of 2HPr is given as a 30% relative error plus the detection limit due to the uncertainty of the cross section determination. Panel (c) - (f) show different model runs for acetaldehyde and methyl glyoxal in which the 2HPr yield was set to 0.66, represented by the solid line in panel (b). As can be seen in panel (c), without considering the 2HPr + OH reaction the simulated profile represents roughly the experimental methyl glyoxal data during the first irradiation period. By contrast, the acetaldehyde profile matches the experimental data only in the beginning of the first irradiation corresponding to a 3P2 consumption of < 30%. The temporal profiles of both species completely fail in reproducing the measured data during the second irradiation, where more than 70% of the 3P2 is already consumed. This demonstrates unambiguously that a secondary source for both acetaldehyde and methyl glyoxal is needed to describe the experimental system, namely the  $\alpha$ hydroxyaldehyde oxidation. However, the match between the simulated and experimental time profiles in the beginning of the first irradiation allows to set values for the first-generation yields of acetaldehyde and methyl glyoxal from 3P2 + OH in these experiments. Panel (d) shows the optimum model run that allows a simultaneous fit of the experimental time profiles for both acetaldehyde and methyl glyoxal. In order to assess the errors for the branching ratios  $k_a$  and  $k_b$ , two scenarios were defined which represent the limiting cases and thus enable to determine the maximum variation of  $k_a$  and  $k_b$ . Accordingly, panel (e) shows a model run in which acetaldehyde is simulated for its lower limit of the accuracy error and methyl glyoxal for the upper limit, respectively, (scenario 1) while panel (f) represents the inverse case (scenario 2). These scenarios were modelled for different strengths of the secondary source of acetaldehyde and methyl glyoxal, means that the 2HPr yield from 3P2 + OH was varied within the limits imposed by the accuracy of the 2HPr measurement, as shown in panel (b). For both scenarios, the obtained first-generation yields of acetaldehyde and methyl glyoxal from 3P2 + OH were found to be independent from the 2HPr yield. Since the formation of acetaldehyde and methyl glyoxal from the 3P2 + OH reaction does not necessarily depend on 2HPr, this observation is rather self-consistent and serves merely as a validation of our model. Based on the proposed 3P2 + OH mechanism one would expect their yields to be the same, thus their ratio should equal unity. This does correspond to scenario 1 while an acetaldehyde/methyl glyoxal ratio > 1 is observed for scenario 2 (Fig. S3 in the Supplement). Although within the accuracy errors this indicates a small bias between the acetaldehyde and methyl glyoxal quantification. The sum of the acetaldehyde and methyl glyoxal yield from 2HPr + OH correlates with the 2HPr yield from the 3P2 + OH reaction, where larger values are observed when the input 2HPr yield is lowered (Fig. S4 in the Supplement). In order to reproduce the entire time profiles of acetaldehyde and methyl glyoxal, an overestimation of the 2HPr mixing ratio and hence the strength of the secondary acetaldehyde and methyl glyoxal source is compensated for by an underestimation of the acetaldehyde and methyl glyoxal yield in the model. Hence, this behaviour can be rationalized in terms of an antagonistic effect. The sum of the acetaldehyde and methyl glyoxal yields becomes unity when a 2HPr yield of about 0.54 and 0.61 is used in the model for scenario 1 and 2, respectively. Considering the yield of 0.66, used to match the experimental data in panel (b), this might indicate an overestimation of the 2HPr mixing ratios. However, the differences are within the accuracy due to the rather uncertain 2HPr cross section. The branching ratios  $k_a$ and  $k_b$  were obtained by scaling of the acetaldehyde and methyl glyoxal yield. These were found to be independent from the 2HPr yield within the 2HPr accuracy limits and almost indistinguishable in between scenario 1 and 2 (Fig. S5 in the Supplement). Accordingly, the average branching ratios  $k_a$  and  $k_b$  are 0.73 ± 0.08 and 0.27 ± 0.08, respectively. Within the uncertainties, this is in agreement with SAR predictions (Kwok and Atkinson, 1995) which estimate 0.8 and 0.2 for the branching ratios, respectively, as well as former results on the OH reaction of glycolaldehyde at atmospheric pressure (Niki et al., 1987; Bacher et al., 2001; Magneron et al., 2005).

Based on these results, the temporal profiles of acetaldehyde and methyl glyoxal are well-reproduced for all conducted experiments. Their corrected yields in the 3P2 + OH reaction are  $0.39 \pm 0.07$  and  $0.32 \pm 0.08$ , respectively. Hence, while larger molar yields were observed for acetaldehyde than for methyl glyoxal without proper corrections the model predicts both first-generation yields to be the same within the accuracy errors, which indicate their formation according to the same reaction channel. The branching ratios of the simplified reaction scheme, obtained through modelling, are given in Tab. 2.

By considering the formation of CH3C(O) radicals from the oxidation of 3P2, acetaldehyde and methyl glyoxal the model underestimates PAN + CO2 at longer reaction times as depicted by the sim1 simulation in Figure 10. This can be partly explained by an additional source of CO2 in the experimental system, since it is also a co-product of acetaldehyde via channel (5a) in the 2HPr + OH reaction (Figure 7). Given that abstraction of the aldehydic H atom of 2HPr is expectedly leading solely to acetaldehyde, the yield of CO2 from 2HPr oxidation depends only on the ratio between decomposition of the hydroxypropionyl radical and its reaction with oxygen (Figure 7).

Figure 8 shows time profiles obtained from an experiment performed in the 480 L chamber, in which PAN and CO2 were quantified, as well as simulated profiles from different model runs. As presented in panel (b), the experimental data are reproduced solely for less than the first half of the irradiation period, if only PAN and CO2 formation from 3P2 + OH are considered in the model. This corresponds to a 3P2 consumption of < 30% which is consistent with the non-linearity of the yield plot observed for higher 3P2 consumption levels (Fig. 3). As discussed before, PAN and  $CO_2$  formation are affected from the further oxidation of acetaldehyde and methyl glyoxal. However, CO2 elimination from the 2HPr derived RO radical (Fig. 6) is an additional source of  $CO_2$  in the experimental system according to pathway (5a). Given that abstraction of the aldehydic H atom of 2HPr is expectedly leading solely to acetaldehyde, the yield of  $CO_2$  from the 2HPr oxidation depends only on the ratio between decomposition of the hydroxypropionyl radical and its reaction with oxygen (Fig. 6). In order to assess the uncertainty on the sum parameter PAN + CO2 due to secondary chemistry, the temporal profile of PAN +  $CO_2$  was simulated assuming both acetaldehyde and methyl glyoxal at the upper (scenario 3) and lower limit (scenario 4) of the measurement accuracy (Fig. 8). Hence, the strength of the secondary sources of CH3C(O) radicals in the experimental system was either maximized or minimized in the model. Moreover, the temporal behaviour of PAN + CO2 was simulated without considering CO2 formation from 2HPr + OH (dashed lines) and assuming the CO2 yield to equal the acetaldehyde yield (solid lines). In both scenarios (panel (d) and (f) of Fig. 8) the temporal profiles are nearly indistinguishable during the first half of the irradiation time and one obtains the same first-generation yield for the sum parameter PAN + CO2, used to determine the  $CH_3C(O)$  radical yield. The entire profile is reproduced solely when the  $CO_2$  yield from 2HPr + OH is equalized to the acetaldehyde yield in scenario 3 (panel (d)). In scenario 4, where the secondary formation of  $CH_3C(O)$  radicals was set to the lower limit, the model slightly underestimates the sum of PAN and  $CO_2$  at the end of the experiment (panel (f)). However, in both scenarios the model predicts the sum of PAN +  $CO_2$  to be significantly lower than experimentally observed at the end of the irradiation period, when the  $CO_2$  formation from 2HPr is set to 0 (dashed lines). When introducing larger PAN +  $CO_2$  yields for 3P2 + OHit is possible to match the observed profile for the second half of the experiment. Although, in this case the model overestimates PAN +  $CO_2$  formation in the first half of the experiment, in which secondary formation is expected to be almost negligible.

For the hydroxyacetyl radical Méreau et al. (2001) concluded, based on *ab initio* calculations, that decomposition cannot compete with the O2 reaction in the case of the structurally similar hydroxyacetyl radical. Niki et al. (1987) observed CO2 instead of CO formation in the glycolaldehyde oxidation when secondary oxidation processes were minimized in the experimental system. These findings together with the significant discrepancy of the simulated and experimental time profile for PAN + CO2 at long irradiation times, when a CO2 formation from 2HPr oxidation is not included in the model, suggest that decomposition of the hydroxypropionyl radical is negligible and  $k_{2a}/[k_{1a} + k_{2a}] = 1$  (see Fig. 6Fig. 7). Including the additional CO2 source in the model improves significantly the consistency between the simulated and experimental PAN + CO2 profile at long irradiation times, although slight discrepancies remain in some experiments As shown in Figure 10 for a 480 L chamber experiment the entire time profile of PAN + CO2 is reproduced when the additional source of carbon dioxide is included into the model. One should note that in this regard the time profile does no longer represent merely the formation of CH3C(O) acetyl radicals. However, given that both the simulation with and without the additional CO2 source are indistinguishable in the first part of the irradiation period (Fig. 8Fig. 10) it is still possible to derive the corrected average yield for PAN + CO2 (0.56 ± 0.14) representing the yield of CH3C(O) radicals.

The lowering of the PAN +  $CO_2$  yield due to the correction is consistent with the presence of secondary processes since both acetaldehyde and methyl glyoxal further oxidation contributes to the  $CH_2C(O)$  radical formation in the experimental system. Besides, as As for acetaldehyde and methyl glyoxal, the yields for 2HPr and PAN +  $CO_2$  are the same within the assigned accuracy thus indicating their formation in the same reaction channel. Since As carbon dioxide formation might be easily affected from processes on the chamber walls and the corrected yield for PAN +  $CO_2$  should, therefore, be still regarded as upper limit. A build-up of  $CO_2$  from the walls might become relevant at longer irradiation times and this supposedly explain the remaining small discrepancies at irradiation times > 10 min in some experiments. However, the reproducibility of the yields without correction is essentially the same as for 2HPr for experiments performed in both chambers. Besides, separate control experiments, in which synthetic air was irradiated with the same set of lamps, did not show significant  $CO_2$  production. Therefore, the influence of off-gasing processes on its temporal behaviour is probably negligible in the beginning of the experiments, when the formation of the products in the target reaction dominates over secondary chemistry. An overestimation of the  $CH_3C(O)$  radical yield is thus unlikely. Uncorrected and corrected molar yields, namely first-generation yields, of all quantified products are summarised in Table 3.

Combining the yields of the 3P2 oxidation products leads to a carbon balance close to unity (0.98 ± 0.18). The branching ratios for the pathways  $\alpha_{ON}$  and  $\beta_{ON}$  (Fig. 4Fig. 5) forming RONO2 species are expectedly very minor channels. This is in agreement with previous work in which RONO2 species from the OH oxidation of  $\alpha$ , $\beta$ -unsaturated ketones were indicated only in our experimental set-up resulting from tertiary RO2-radicals (Illmann et al., 2021b). This is in agreement with previous findings in our laboratory, where the production of RONO2 species in the OH oxidation of  $\alpha$ , $\beta$ -unsaturated ketones was observed only in conjunction with the formation of tertiary RO2 radicals (Illmann et al., 2021b). FurtherBesides, Praske et al. (2015) reported a low overall RONO2 yield of 0.040 ± 0.006 for MVK oxidation."

The Figs. 7 and 8 (formerly 9 and 10) were replaced as follows: